# Distinct roles of *Atf3*, *Zfp711* and *Bcl6b* in early embryonic hematopoietic and endothelial lineage specification

Ridvan Cetin[1], Giulia Picco[1], Jente van Staalduinen[1], Eric Bindels[2], Remco Hoogenboezem[2], Gregory van Beek[2], Mathijs A. Sanders[2], Yaren Fidan[1], Ahmet Korkmaz[3], Joost Gribnau[1,4], Jeffrey van Haren[1], Danny Huylebroeck[1], Eskeatnaf Mulugeta[1] and Frank Grosveld[1,*]

## ABSTRACT

Hematopoiesis occurs in three consecutive overlapping waves in mammals, regulated by transcription factors. We investigated the role of three relatively poorly studied transcription factors in early embryonic hematopoietic development at single-cell resolution: *Atf3*, *Zfp711* and *Bcl6b*. These transcription factors are upregulated early in development, when hematopoietic and endothelial lineages separate from cardiac and other mesodermal lineages. We combined multiplexed single-cell RNA sequencing and flow cytometric analysis with knockouts in *in vitro* differentiating mouse embryonic stem cells to dissect the function of these transcription factors in lineage specification. Δ*Atf3* cells showed increased mesodermal differentiation but decreased endothelial cells and erythro-myeloid progenitors, accompanied by aberrant interferon signaling. Mechanistically, loss of *Atf3* disrupted key hematopoietic regulatory genes (*Runx1*, *Egr1*, *Jun*, *Fos*, *Mafb* and *Batf3*) required for the formation of erythro-myeloid progenitors. Δ*Zfp711* cells exhibited increased blood progenitors and erythroid cells, but decreased endothelial cells, with a striking shift from *Hoxa*+ mesoderm (allantois and limb mesoderm) to *Hoxb*+ mesoderm (mesenchyme and epicardium). Notably, *Zfp711* binds the *Atf3* promoter, suggesting a hierarchical regulation. In contrast, Δ*Bcl6b* had no observable effects on early hematopoiesis, despite specific expression in hemato-endothelial progenitors.

KEY WORDS: **Atf3, Bcl6b, Zfp711, Hematoendothelial development, Erythro-myeloid progenitors, Endothelial-to-hematopoietic transition**

## INTRODUCTION

Blood vessel and blood cell formation are essential for mammalian embryonic development beyond the diffusion-limited early stages. These processes occur in three overlapping hematopoietic waves, beginning with hemato-endothelial progenitors (HEPs) that generate primitive erythrocytes, megakaryocytes and endothelium at embryonic day (E)7.5 in yolk sac blood islands, followed by erythro-myeloid progenitors (EMPs) arising through endothelial-to-hematopoietic transition (EHT) in the second wave, and finally hematopoietic stem cells emerging from hemogenic endothelium of the dorsal aorta (E10.5) to sustain lifelong hematopoiesis (Dzierzak and Bigas, 2018; Harland et al., 2021; Neo et al., 2021; Tober et al., 2006; Yamane, 2020). The precise regulation of these waves is crucial, as disruption leads to embryonic lethality or severe hematopoietic disorders (Vink et al., 2022).

Understanding transcription factor (TF) regulation of these waves is crucial to generating hematopoietic cells *in vitro*, potentially reducing dependence on blood donors and transplantation complications. Current protocols struggle to efficiently generate functional HSCs *in vitro*, highlighting gaps in our understanding of the transcriptional networks governing hematopoietic specification (Sugimura et al., 2017). Recent advances in single-cell RNA sequencing (scRNA-seq), directed differentiation protocols and integrated genomic databases now enable investigation of TF functions, networks and their cooperative functions at single-cell resolution during development (Haniffa et al., 2021; Imaz-Rosshandler et al., 2023; Karlsson et al., 2021; Pijuan-Sala et al., 2019). Additionally, directed differentiation protocols using pluripotent stem cells now successfully recapitulate *in vivo* developmental processes across multiple lineages, including hematopoietic (Ng et al., 2025), extraembryonic mesodermal (Theeuwes et al., 2025), neural (He et al., 2024; Hong and Do, 2019) and endodermal (Xu et al., 2025) differentiation.

While master regulators like GATA1 and RUNX1 are well characterized (De Bruijn and Dzierzak, 2017), the roles of many TFs expressed during hemato-endothelial divergence remain poorly understood. Using mouse embryonic stem cell (mESC) differentiation, we identified three TF genes upregulated when hemato-endothelial lineages diverge from cardiac mesoderm at day 4 (D4): activating transcription factor 3 (Atf3), zinc-finger protein 711 (Zfp711) (the human *ZNF711* ortholog) and B cell CLL/lymphoma 6 member B (Bcl6b). ATF3, a bZIP domain-containing factor, forms repressive homodimers or activating heterodimers with JUN family members (Chen et al., 1994; Hai et al., 1999; Liang et al., 1996). ZFP711 contains C2H2 zinc fingers, binds CpG islands and recruits histone demethylases to regulate transcription (Rhie et al., 2018; Wu et al., 2021). BCL6B functions as a transcriptional repressor through zinc finger-mediated DNA binding, requiring dimerization with BCL6 (Takamori et al., 2004).

Here, we address whether these TFs regulate: (1) early hemato-endothelial versus mesodermal lineage decisions; (2) later differentiation stages, including EHT and primitive hematopoiesis; (3) which genes and pathways change upon their respective perturbation; and (4) whether multiplexed scRNA-seq with flow

[1]Former Department of Cell Biology, Erasmus University Medical Center Rotterdam, Rotterdam 3000 CA, The Netherlands. [2]Department of Hematology, Erasmus University Medical Center Rotterdam, Rotterdam 3000 CA, The Netherlands. [3]Medical Faculty, Institute of Physiology, RWTH Aachen University, Aachen 52074, Germany. [4]Department of Developmental Biology, Erasmus University Medical Center Rotterdam, Rotterdam 3000 CA, The Netherlands.

*Author for correspondence (f.grosveld@erasmusmc.nl)

R.C., 0009-0000-5743-8888; G.P., 0009-0002-6370-1480; J.v.S., 0000-0001-6790-8345; E.B., 0000-0001-9502-669X; R.H., 0000-0002-1719-6455; G.v.B., 0000-0003-0283-7069; M.A.S., 0000-0002-8575-9213; Y.F., 0009-0003-9395-1461; J.G., 0000-0001-5645-4691; J.v.H., 0000-0002-3160-3547; D.H., 0000-0003-4862-1079; E.M., 0000-0003-4045-4835; F.G., 0000-0002-7051-4715

cytometry provides efficient, parallel analysis of multiple TFs that is cost-effective and batch-effect-free.

Our integrated approach reveals that *Atf3* plays a role in EMP abundance and/or maturation, endothelium abundance and the generation of various new cell types in late mesodermal lineages, some of which show strong expression of interferon pathway-related genes; *Zfp711* affects mesodermal cell type abundance and *Atf3* is one of the genes affected by loss of *Zfp711*; *Bcl6b* shows no observable impact on hemato-endothelial lineages or mesodermal development. These findings highlight the strength of the adapted workflow to dissect TF functions in hemato-endothelial lineage specification in a cost-effective and batch-effect-free way. Understanding these regulatory mechanisms has implications for *in vitro* hematopoietic differentiation protocols and developing novel therapeutic strategies for blood disorders.

## RESULTS

### Combined multiplexed scRNA-seq and flow cytometry approach for parallel TF knockout analysis

We developed a multiplexed approach for three genes, combining scRNA-seq with flow cytometric analysis (FCA) (Fig. 1A-C) to provide greater depth than large-scale multi-gene CRISPR screens (Adamson et al., 2016; Datlinger et al., 2017; Dixit et al., 2016; Jaitin et al., 2016), which is more efficient than traditional single-gene studies (Harland et al., 2021). We investigated these three TFs by generating CRISPR-Cas9 knockout (KO) mESCs for *Atf3*, *Bcl6b* and *Zfp711* (Δ*Atf3*/*Atf3*-KO, Δ*Bcl6b*/*Bcl6b*-KO and Δ*Zfp711*/*Zfp711*-KO) based on their increased expression during hemato-endothelial specification both *in vitro* and *in vivo* (Fig. 1D,E; Figs S1, S2).

The KO lines and non-targeted controls (NT-Controls) were differentiated using established protocols, with three biological replicates multiplexed per experiment using cell multiplexing oligonucleotides (CMOs). This design captured 71,107 cells across five libraries (16,404 to 18,948 cells per condition), providing sufficient depth to detect changes in rare populations while minimizing batch effects and preserving biological variation without requiring computational integration (Fig. S3).

For orthogonal validation, we performed FCA at two key timepoints: D4 and D7. At D4, we assessed HEPs (Flk1+/Pdgfra−) versus other mesodermal lineages (Flk1+/Pdgfra+). At D7, we employed a seven-marker panel to identify populations from the first and second hematopoietic waves: blood progenitors (CD41/encoded by *Itga2b* and CD71/*Tfrc*), endothelium (CD144/*Cdh5* and CD102/*Icam2*), EMP (CD45/*Ptprc*) and late mesodermal lineages (CD140a/*Pdgfra* and CD140b/*Pdgfrb*) (Fig. 1C; Fig. S4) (Harland et al., 2021; Pijuan-Sala et al., 2019; Psaila et al., 2016; Sievert et al., 2014). The scRNA-seq was performed at D7, capturing cell types from the first and second waves of hematopoiesis simultaneously. Notably, CD45 is a specific marker for this differentiation window (D4-D7; Fig. S5); later timepoints or alternative differentiation protocols require modified marker panels (McGrath et al., 2015).

### Cell type identification reveals a complete and batch effect-free hemato-endothelial differentiation trajectory

The scRNA-seq results showed no significant differences in cell distributions between the libraries and conditions, indicating no need for computational batch correction and integration methods (Fig. S3C-E). The scRNA-seq data show a continuous differentiation pattern, starting from naive pluripotency mESCs to multiple differentiated cellular states (e.g. erythroid, EMP, endothelium, megakaryocytes, amniotic ectoderm, visceral endoderm and other late mesodermal lineages) (Fig. 2A). The cell subpopulations were

analyzed at three hierarchical levels, i.e. 'groups', 'clusters' and 'sub-clusters' (Fig. 2B,C). There are three groups – early differentiation, hematopoietic and late mesoderm – subdivided into 15 clusters (Fig. 2B) and 40 sub-clusters (Fig. 2C) by virtue of highly expressed marker genes (Figs S6-S10).

### ATF3

#### *Atf3* deletion impairs endothelial and EMP differentiation

We first analyzed the expression of *Atf3* and noted an (apparent) contradiction between the *in vitro* and *in vivo* data (Imaz-Rosshandler et al., 2023). *Atf3* is not expressed in EMP#1 and is upregulated in EMP#2, EMP#3 and EMP#4, while it is not expressed in the *in vivo* data (Figs S11 and S12). This is because the *in vivo* data represent only the early stage of EMP maturation, as demonstrated by the expression of *Kit*, *Ctla2a* and *Nrgn*, and other genes are expressed in EMP#1 and turned off in EMP#2, EMP#3 and EMP#4, in agreement with the *in vivo* data (Fig. S11). We therefore conclude that the *in vivo* data represent an early stage of EMP differentiation. This difference is not seen in the other tissues (Fig. S12), which show an upregulation of *Atf3* (allantois, epicardium, mesenchyme and endothelium). A similar apparent difference in EMP maturation is seen for *Zfp711* (see below).

CRISPR-mediated deletion of *Atf3* (8 kb spanning exons 2-4) eliminated the *Atf3* expression (Fig. S11A) that is normally seen in *in vitro* and *in vivo* differentiation (Figs S11-S13). Flow cytometry at D4 showed increased Pdgfra+ mesodermal cells (11.8%), while HEPs remained unchanged. By D7, mesodermal markers further increased (Pdgfra+, 10.6%; Pdgfrb+, 13%; double positive, 28.7%), but endothelial cells (CD144/*Cdh5*+CD102/*Icam2*+) decreased by 41% and CD45+ EMPs by 22% compared to controls; erythroid differentiation (CD41+CD71+) was unaffected (Fig. 3; Table S1A,B).

Single-cell analysis confirmed these findings through differential abundance analysis (DAA), revealing a decrease in mature EMPs (EMP#3 and EMP#4) but an increase in immature EMPs (EMP#1), suggesting disrupted EHT progression (Fig. 4; Figs S14 and S15; Tables S2 and S3). A novel Δ*Atf3*-enriched population emerged, characterized by high interferon-responsive gene expression (*Ifit1*, *Rsad2* and *Stat1*).

### Transcriptional changes reveal dual *Atf3* functions

Differential gene expression analysis (DGEA) identified 124 differentially expressed genes (DEGs) in EMPs (107 downregulated, 17 upregulated) and 89 in late mesoderm (69 upregulated, 20 downregulated), with mature EMPs showing the most changes (150 DEGs) (Fig. 5A,B; Table 1; Fig. S16A,B; Table S4A-H). Integration with ChIP-seq and ATAC-seq data categorized 223 genes as potential direct Atf3 targets based on promoter proximity (−300 bp to +125 bp from TSS). Cross-referencing with ENCODE and published knockout data identified 63 high-confidence direct targets, including *Egr1*, *Jun*, *Fos*, *Batf3* and *Atf3* itself (Fig. 5C-F; Table 2; Fig. S17; Table S7A-E).

The Δ*Atf3*-enriched and endothelium#3 populations shared 108 upregulated interferon-response genes (*Ifit1*, *Ifit2*, *Ifit3*, *Ifitm3* and *Bst2*), representing 51 of the most significant DEGs, indicating that *Atf3* normally suppresses interferon signaling during hematopoietic specification (Fig. 5G,H; Fig. S16C-F).

### Gene set enrichment reveals pathway-level changes

Gene set enrichment analysis (GSEA) revealed contrasting effects in EMPs versus late mesoderm. In EMPs, cell cycle and RNA-processing pathways were enriched, while migration, endocytosis and myeloid functions were suppressed (Fig. S18A-C; Fig. S19A-C, Fig. S20A; Table S5A-C). Late mesoderm showed opposite patterns,

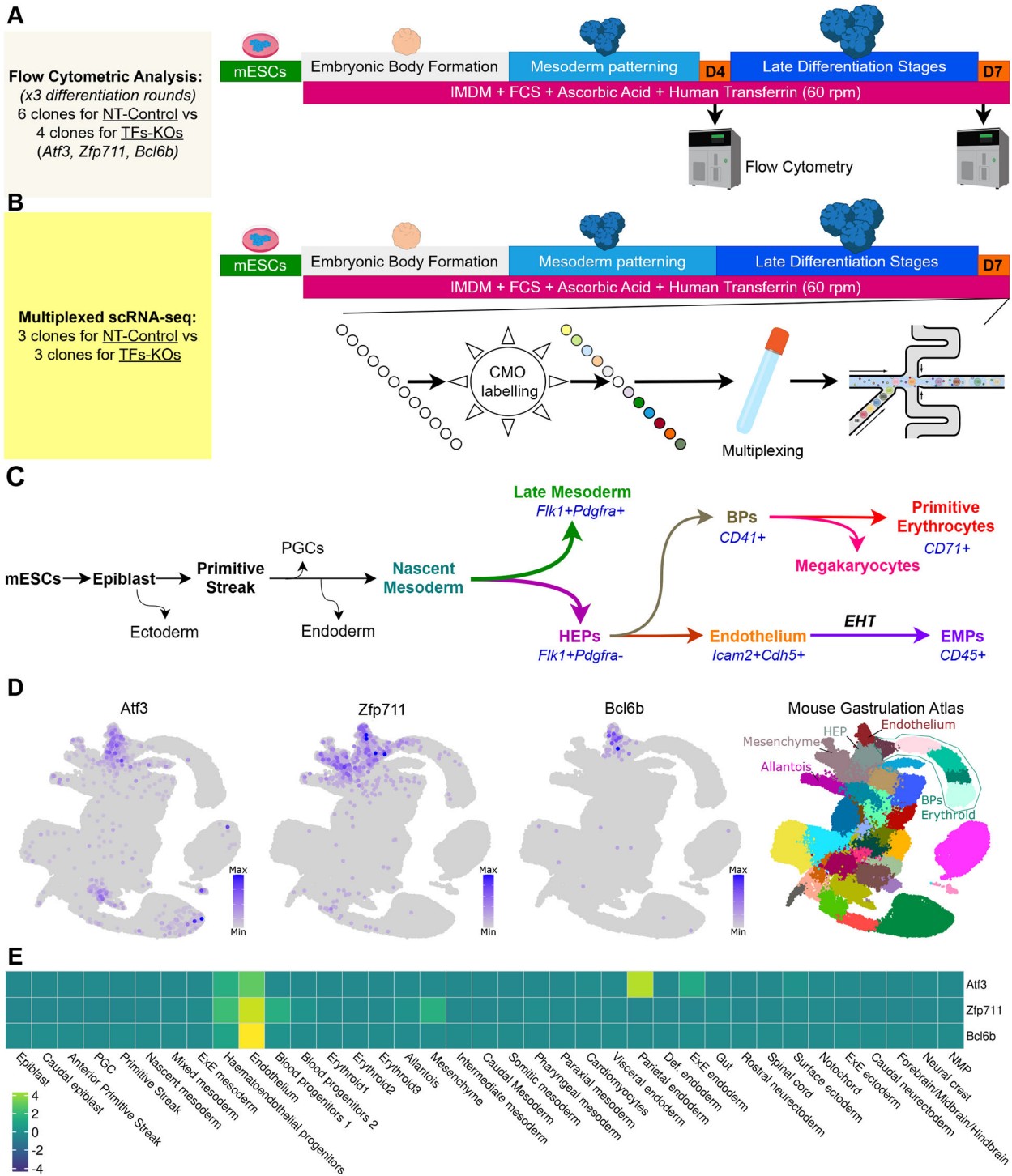

**Fig. 1. Experimental design and transcription factor expression patterns during hemato-endothelial development.** (A) Flow cytometric analysis (FCA) workflow for assessing differentiation of knockout and control mESC lines. Experiments were performed using four independent knockout clones and six control clones per condition, analyzed at days 4 (D4) and 7 (D7) of differentiation, and the whole process repeated three times in different differentiation rounds. (B) Multiplexed scRNA-seq experimental design showing simultaneous analysis of three knockout lines and controls at D7, with three biological replicates per condition processed in a single batch using cell multiplexing oligonucleotides (CMOs). (C) Schematic of mesodermal and hematopoietic lineage trajectories showing the temporal emergence of cell populations and corresponding surface markers used for FCA. (D,E) *In vivo* expression patterns of *Atf3*, *Zfp711* and *Bcl6b* during mouse embryonic development (E6.5-E8.5). Data in D and E were produced using the mouse gastrulation atlas (Pijuan-Sala et al., 2019) showing cell-type-specific expression.

with increased interferon signaling ($\alpha$ and $\gamma$), TGF$\beta$ and NF-$\kappa$B pathways, but decreased proliferation (Fig. S18D-F; Figs S19D-F, S20B). These opposing profiles suggest *Atf3* promotes EMP-specific programs while restraining interferon responses in mesoderm.

**Atf3 cooperates with hematopoietic factors during EHT**

ATAC-seq analysis revealed dynamic chromatin remodeling at the *Atf3* and other loci during EHT, with multiple new enhancer peaks emerging in EMPs (Fig. S21A). We then looked at the sequence of

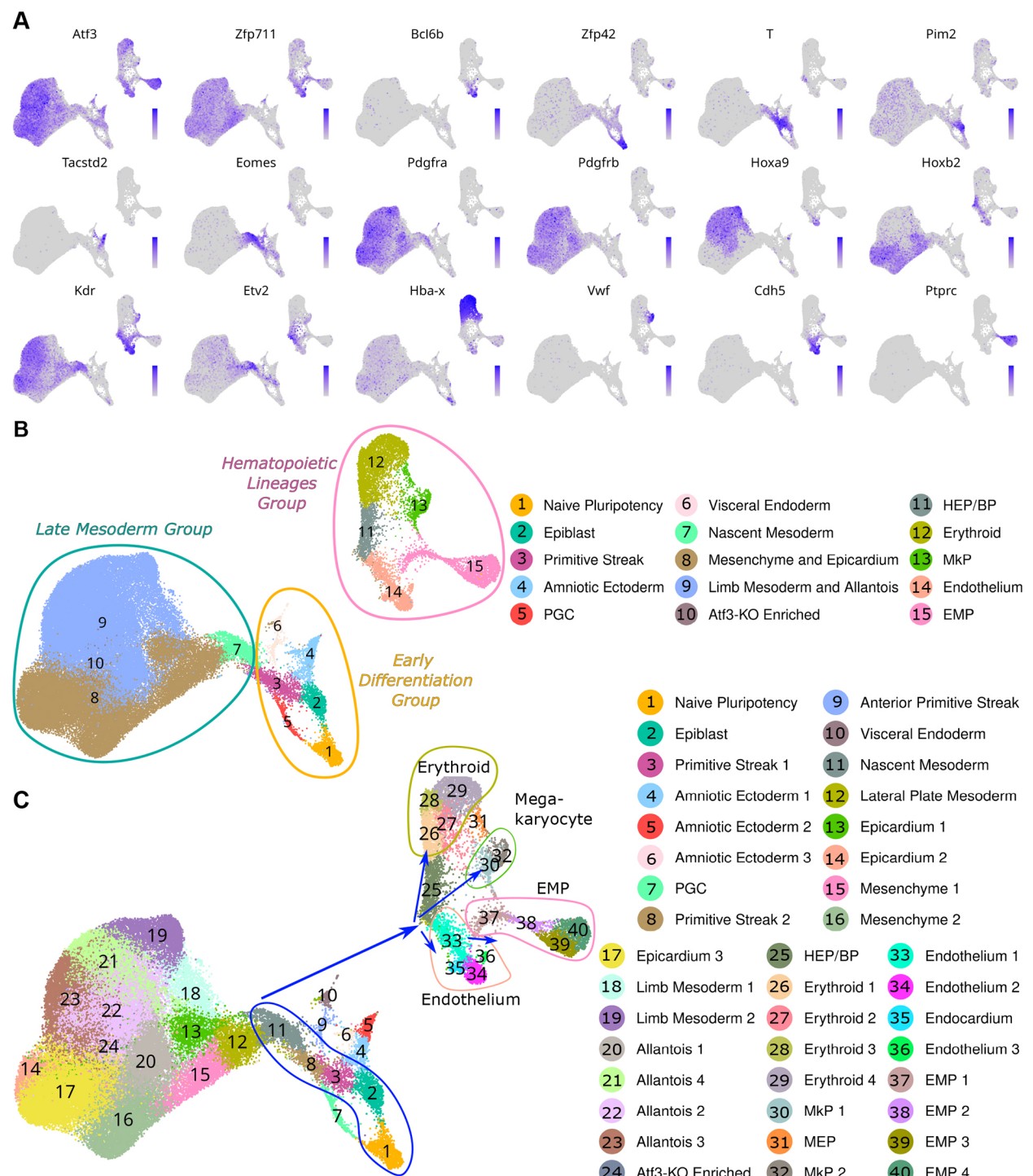

**Fig. 2. Identification of cell groups, and clusters and sub-clusters therein, based on marker gene expression in scRNA-seq results of D7 embryoid bodies.** (A) UMAP projections showing expression patterns of the three studied transcription factors (*Atf3*, *Zfp711* and *Bcl6b*) and lineage-defining marker genes. *Zfp42* marks naive pluripotent cells, *T* and *Pim2* mark epiblast and primitive streak, *Eomes* marks early/nascent mesoderm differentiation, *Pdgfra* and/ or *Pdgfrb* mark mesoderm, *Hoxa9* and/or *Hoxb2* mark patterned mesoderm, *Kdr* and *Etv2* mark hemato-endothelial progenitors, *Hba-x* marks erythroid cells, *Vwf* marks megakaryocytes, *Cdh5* marks endothelium, and *Ptprc* marks EMPs. Expression levels are indicated by color intensity (grey, low; blue/purple, high). (B) Hierarchical cell type organization showing three major groups (group level) and 15 clusters (cluster level). Early differentiation group (orange outline) contains naive pluripotent and early differentiation populations. Hematopoietic lineages group (pink outline) includes blood progenitors, erythroid cells, megakaryocytes, endothelium and EMPs. Late mesoderm group (turquoise outline) comprises Hox-patterned mesoderm and its derivatives. The Δ*Atf3*-enriched cluster (10) appears uniquely in knockout cells. (C) Fine-resolution analysis identifying 40 sub-clusters (subcluster level) numbered by differentiation maturity within each lineage. Blue trajectory arrows indicate the developmental progression from naive pluripotent cells (1) through mesoderm specification (2, 3, 8 and 11) to terminal differentiation into erythroid (26-29), megakaryocyte (30,32), endothelial (33-36) and EMP (37-40) lineages. Sub-cluster annotation combines lineage identity with maturation stage (e.g. Erythroid 1-Erythroid 4, EMP 1-EMP4).

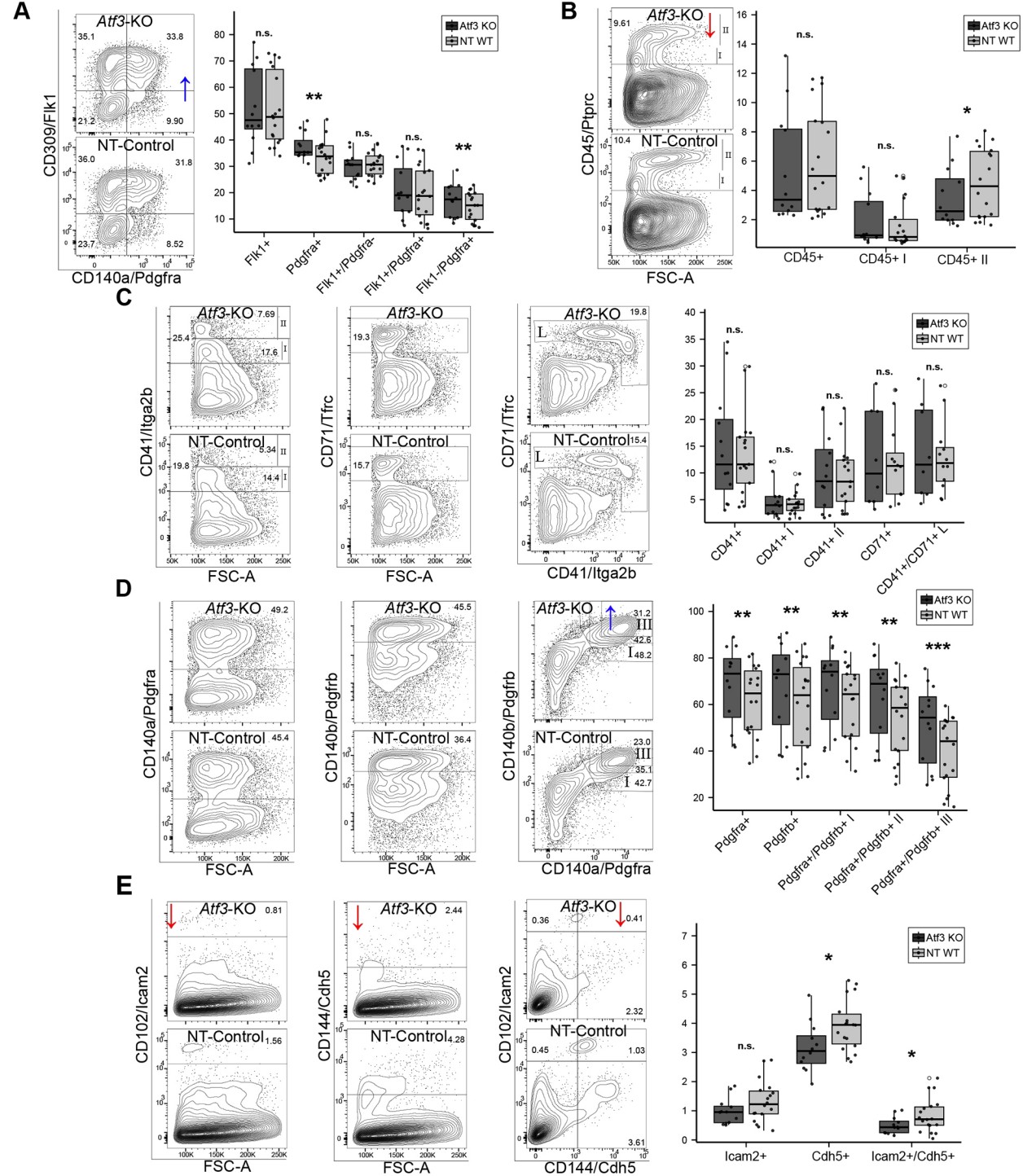

**Fig. 3. Flow cytometric validation of *Atf3* knockout effects on mesodermal and hemato-endothelial differentiation.** (A) D4 analysis showing distribution of hemato-endothelial progenitors (Flk1$^+$/Pdgfra$^-$) and mesodermal populations (Flk1$^+$/Pdgfra$^+$) in Δ*Atf3* versus NT-Control cells. Representative plots (left) and quantification (right) demonstrate increased Pdgfra$^+$ populations. (B) D7 EMP analysis using CD45 marker. Representative plots and quantification show reduction in CD45$^+$ EMPs in Δ*Atf3*. (C) D7 erythroid lineage assessment using CD41 and CD71 markers. No significant differences observed between Δ*Atf3* and controls. (D) D7 late mesoderm populations marked by Pdgfra and/or Pdgfrb. Significant increases in single and double-positive populations in Δ*Atf3*. (E) D7 analysis of endothelial markers. Representative plots and quantification show reduction in Cdh5$^+$/Icam2$^+$ endothelial cells in Δ*Atf3*. Boxes indicate the interquartile range (IQR; 25th-75th percentile), the central line indicates the median, whiskers extend to 1.5× IQR, and points beyond represent outliers. Data are from three independent experiments with four knockout and six control clones. *$P<0.05$, **$P<0.01$, ***$P<0.001$; n.s., not significant.

these peaks and carried out a motif analysis to determine which other TFs are bound close to ATF3 in the same peaks. This yielded a number of TFs with Spi1 as the top candidate binding next to

ATF3. Very similar observations were made for these and a number of other transcription factors (Fig. S21B,C). Analysis of the expression patterns of *Spi1* and *Cebpb* showed that these

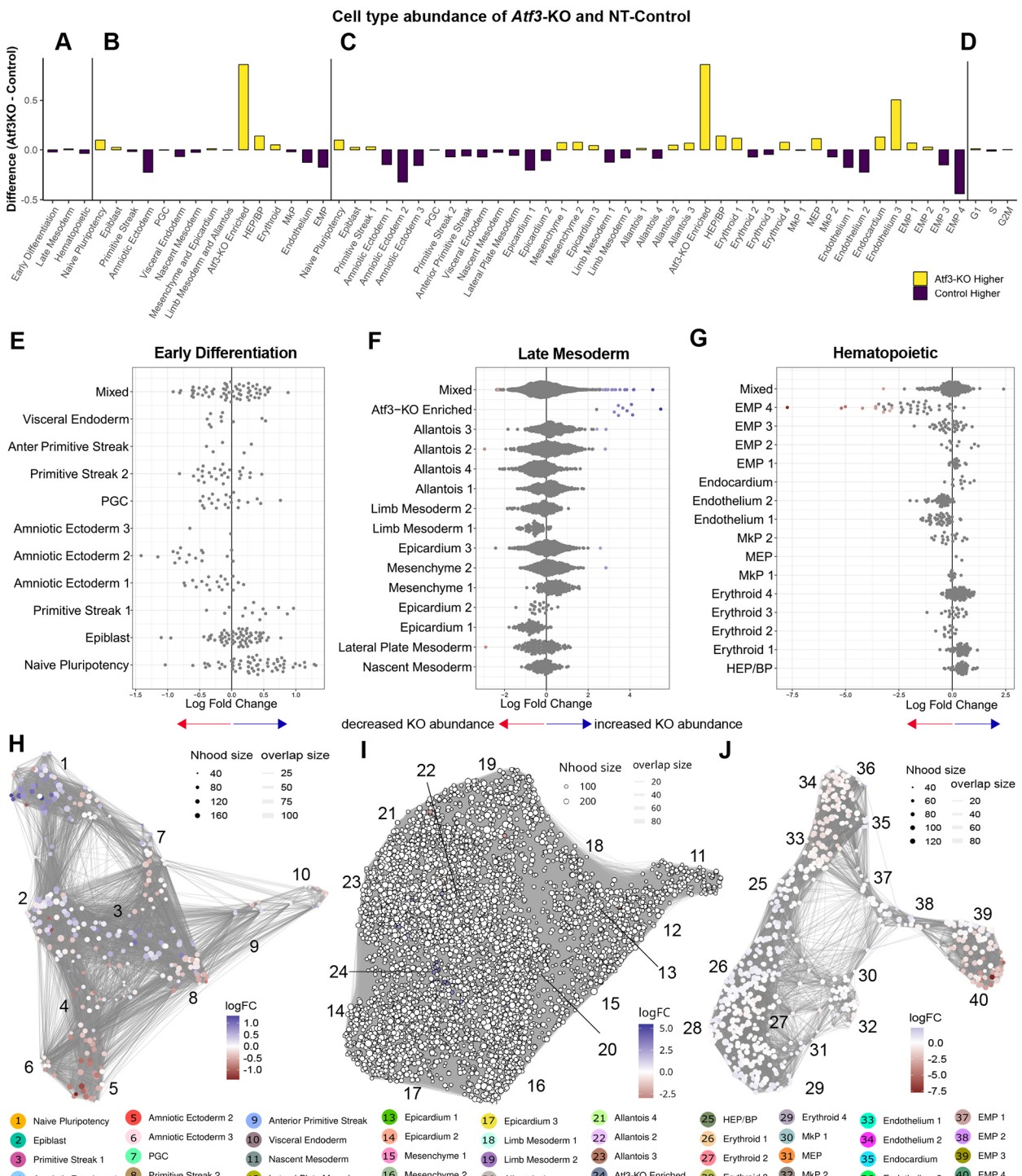

**Fig. 4. Single-cell differential abundance analysis reveals *Atf3*-dependent changes in cell population dynamics.** (A-D) Cell type abundance differences between Δ*Atf3* and NT-Control across hierarchical annotations. Yellow bars indicate higher abundance in Δ*Atf3*; purple bars indicate lower abundance. (A) Group level, (B) cluster level, (C) sub-cluster level and (D) cell cycle phases. Proportions of the cell types produced with Speckle/Propeller. (E-G) MiloR k-nearest neighbor analysis showing log fold-changes in cell abundance by group with subcluster annotation. (E) Early differentiation populations, (F) late mesoderm and (G) hematopoietic lineages (Spatial FDR<0.1). (H-J) Neighborhood graph visualization of differential abundance produced with MiloR. (H-J) Cluster-level network showing relationships between Early differentiation, late mesoderm and hematopoietic populations. UMAP with neighborhood overlay (node size=neighborhood size). Color scale indicates log fold-change (red, decreased; blue, increased in Δ*Atf3*).

factors are newly expressed in the transition from endothelium to EMP. We then looked at the TFs in our DEGs going from endothelium to EMP and confirmed that all of these genes had ATF3 Chip-seq peaks as expected. We therefore suggest that the previously unreported expression of hematopoietic TFs in endothelium together with ATF3 results in differentiation to EMP and that this EHT process is delayed by the absence of Atf3.

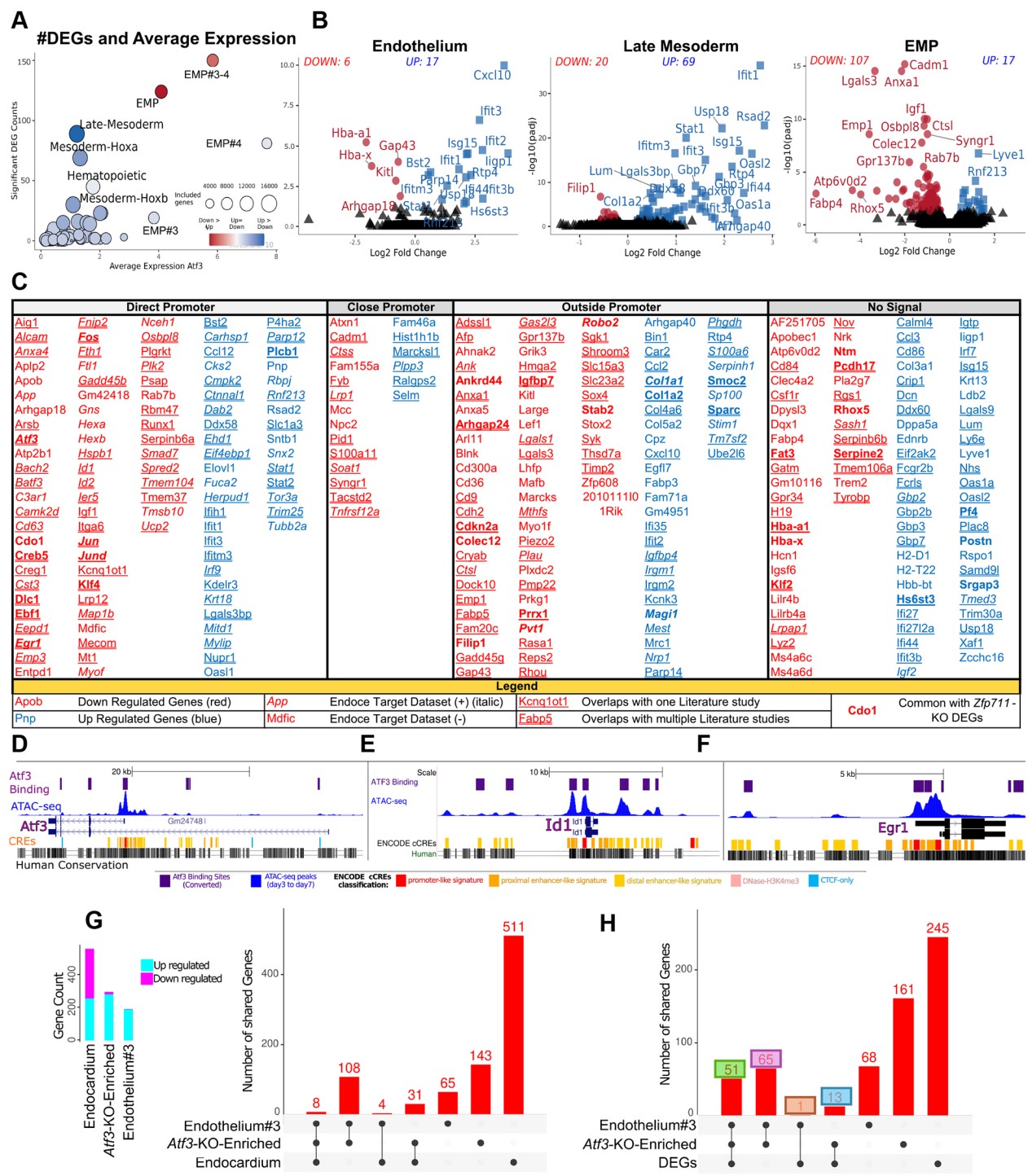

**Fig. 5. Results and categorization of the DGEA of the ΔAtf3 cells.** (A) Differential gene expression (adjusted-*P*<0.05) across hierarchical cell annotations. Circle size represents genes analyzed after quality filtering; color indicates upregulation (blue) or downregulation (red). Mature EMPs (EMP#3 and EMP#4) show the highest DEG count despite smaller cell numbers, indicating biological significance. (B) Volcano plots revealing contrasting transcriptional responses: downregulation in endothelium and/or EMPs versus upregulation of interferon-response genes (*Ifit1*, *Isg15* and *Rsad2*) in late mesoderm. (C) Comprehensive DEG categorization integrating promoter proximity (−300 bp to +125 bp=direct promoter), ATAC-seq accessibility, ENCODE targets and published *Atf3* studies. A single underlined gene shows that the gene is shared with DEGs from other studies in the literature once and a double underlined gene shows that the gene is shared with DEGs from other studies in the literature more than once. (D-F) Representative genomic tracks showing *Atf3* binding sites coinciding with ATAC-seq peaks at target genes. (D) *Atf3*, (E) *Id1* and (F) *Egr1*. (G) Venn analysis of marker genes from emerging ΔAtf3-enriched populations showing 108 shared interferon-response genes between Endothelium#3 and ΔAtf3-enriched mesoderm. (H) Overlap between population-specific markers and total ΔAtf3 DEGs. The colored boxes refer to the number of intersected genes shown in Fig. S16F.

In summary, lack of Atf3 showed the strongest phenotype among the three TFs, and showed increased and high expression in EMPs, which increases with maturation (EMP#1 to EMP#4). Loss of Atf3 caused dual effects: increased mesodermal differentiation (Pdgfra$^+$/Pdgfrb$^+$ cells) with aberrant interferon signaling; and impaired hemato-endothelial development (41% fewer endothelial cells,

**Table 1. ΔAtf3 cells: selected DEGs (TFs and other important genes)**

| | |
|---|---|
| Late mesoderm | *Stat1, Stat2, Irf9, Irf7, Id1, Zfp608, Stox2, Jun, Jund, Klf4, Smad7, Runx1, Prrx1, Trim25, Xaf1, Eif2ak2*, **Npas3**, **Id3** and **Trps1** |
| *Hoxa*⁺ mesoderm | *Stat1, Ebf1, Id1, Irf9, Klf4, Creb5, Bach2, Smad7, Stox2, Trim25, Xaf1, Zfp608, Rhou, Id3* and *Hand1* |
| *Hoxb*⁺ mesoderm | *Stat1, Irf9, Mecom, Trim25*, **Id1**, **Zf608** and **Smad7** |
| Hematopoietic | *Irf7, Stat1, Batf3* and *Blnk* |
| EMP | *Id2, Mafb, Rhox5, Atxn1, Fos, Irf7, Blnk, Gadd45b, Gadd45g, Ldb2, Sp100*, **Zeb2**, **Xaf1** and **Nov** |
| EMP#4+EMP#3 | *Id2, Mafb, Irf7, Atxn1, Sox4, Creg1, Sash1, Gadd45b, Blnk*, **Fos**, **Hmga2**, **Syk**, **Xaf1** and **Rbm47** |

Adjusted-*P* values are between 0.05 and 0.1 for the genes in bold, and <0.05 for the genes in italic.

22% fewer EMPs). ATAC-seq revealed that *Atf3* cooperates with EHT-specific factors (Spi1 and Cebpb) at enhancers emerging during the EHT. Without Atf3, this regulatory network is disrupted by effecting expression of the important TF genes for EHT (e.g. *Runx1*, *Atf3*, *Cebpb*, *Egr1*, *Batf3*, *Fos* and *Fosb*).

### ZFP711

#### *Zfp711* deletion enhances erythropoiesis while reducing endothelial differentiation

CRISPR-mediated deletion of *Zfp711* (19 kb spanning exon 3 to terminal exon) deleted DNA-binding domains, which eliminated the functional expression normally seen *in vitro* and *in vivo* differentiation (Figs S22-S24; Table S9B). Flow cytometry at D4 showed modest increases in mesodermal populations (Flk1⁺Pdgfra⁺, 12%; HEPs, 4.8%). By D7, changes were pronounced: Pdgfra⁺Pdgfrb⁺ mesoderm increased by 9.9%, while Cdh5⁺Icam2⁺ endothelium decreased by 43%. Remarkably, erythroid markers increased substantially (CD41, 16%; CD71, 24%) (Fig. 6; Table S1C,D).

Single-cell analysis confirmed the Hox-patterning shift, with all limb mesoderm-allantois (Hoxa⁺) sub-clusters decreasing (22-33%) and mesenchyme-epicardium (Hoxb⁺) sub-clusters correspondingly increasing. Blood progenitors nearly doubled (89% increase), mature erythroid cells increased by 45%, while some EMP sub-clusters decreased (Fig. 7; Figs S25, S26; Tables S2I-P, S3D-F).

#### Transcriptional analysis reveals *Zfp711-Atf3* regulatory hierarchy

Differential expression analysis identified 203 DEGs, concentrated in late mesoderm (139 DEGs: 96 down, 43 up) with minimal changes in hematopoietic clusters (Fig. 8A,B; Table S4L-T). Limb mesoderm-allantois (Hoxa⁺) mesoderm specifically lost mRNA expression of transcription factors (*Atf3, Klf4, Klf6, Klf9, Hes1, Jund, Fosb* and *Zswim6*) and growth factors (*Dusp1* and *Hbegf*), while mesenchyme-epicardium (Hoxb⁺) mesoderm showed distinct changes (*Klf2, Asb4* and *Robo2* decreased; *Tox3* and *Adgrb3* increased) (Fig. 8B, Table 3; Tables S4L-T, S10C,D).

Most Zpf711-binding sites are located downstream of a TSS (defined here as up to +500 bp; Fig. S27A,B). The DEGs again fell into four discernable categories (Tables S7F-J, S4L): 'promoter' and 'close promoter', 'distant promoter' and 'no signal' (Fig. 8D-F, Table 4). The DEGs of published ZNF711 knockout and knockdown studies overlapped with our DEGs (Fig. 8D; Fig. S27; Table S6B). Interestingly, *Atf3* was among the DEGs and has a Zfp711-binding site in its promoter (Fig. 8B,F). The DEG lists of the ΔZfp711 and ΔAtf3 differentiated mESCs shared 40 genes (30 of which are upregulated, 10 of which are downregulated), including *Atf3* itself (Fig. 8C).

#### Pathway analysis confirms effects distinct from *Atf3*

GSEA of late mesoderm revealed decreased inflammatory signaling (TNFα/NF-κB, interferon-γ), and increased RNA processing and intracellular transport (Figs S28, S29; Table S5G-I).

**Table 2. ΔAtf3 cells: selected DEGs and neighboring Atf3 peaks with GREAT single nearest gene assignment**

| Gene | Single nearest gene assigned *Atf3* peaks | Closest upstream | Closest downstream |
|---|---|---|---|
| *Atf3* | −58,902, −41,164, −32,983, −11,035, −10,620, −61, +6021 | −61 | +6021 |
| *Batf3* | −240, +389, +20,928, +26,001, +27,766 | −240 | +389 |
| *Bach2* | −98,268, −89,399, −78,049, −58,772, −113, +4901, +8173, +11,188, +11,842, +12,244, +49,292, +65,051, +113,009, +143,287 | −113 | +4901 |
| *Egr1* | −9534, +188, +719, +1546, +5379, +5880, +24,231 | −9534 | +188 |
| *Fos* | −39,538, −21,995, −18,536, −1705, −17, +5410, +9190, +12,089, +49,960 | −17 | +5410 |
| *Gadd45b* | −1283, +73, +9838, +10,473, +31,206 | −1283 | +73 |
| *Id1* | −9388, −9015, −6644, −1058, +41, +3457, +5396, +6345 | −1058 | +41 |
| *Id2* | −378,152, −290,567, −238,663, −213,545, −208,242, −206,604, −154,428, −112,481, −92,353, −86,251, −48,263, −43,544, −42,979, −35,033, −13,349, −3947, −1673, +1124, +4095, +4718 | −1673 | +1124 |
| *Ier5* | +66, +1351, +10,161, +10,897, +12,621, +24,493, +39,936, +49,318, +59,001, +60,233, +94,119 | | +66 |
| *Irf9* | +656 | | +656 |
| *Jun* | −178,989, −160,830, −156,659, −155,432, −136,147, −109,364, −104,761, −91,739, −53,465, −29,813, −29,447, −23,937, −16,153, −1088, −75, +2022, +3572, +26,059, +36,259 | −75 | +2022 |
| *Jund* | −154, +1709 | −154 | +1709 |
| *Klf4* | −581,490, −536,538, −485,877, −258,128, −201,114, −119,113, −60,483, −39,167, −9368, −288, +9364, +54,392 | −288 | +9364 |
| *Mafb* | −8918, +17,224 | −8918 | +17,224 |
| *Mecom* | −2479, −201, +122,179 | −201 | +122,179 |
| *Runx1* | −371,400, −367,518, −335,397, −257,673, −237,757, −180,801, −153,709, −137,645, −126,431, −57,876, −2156, −118, +24,380, +31,760, +39,015, +85,770, +110,415, +127,287, +128,925, +150,235, +150,829 | −118 | +24,380 |
| *Smad7* | −105,009, −92,923, −67,255, −47,981, −45,186, −6773, −735, −222, +8071, +17,365, +18,917, +21,947, +24,271, +30,141, +38,240, +41,380, +63,329, +77,156, +87,766, +127,809, +135,494 | −222 | +8071 |
| *Stat1* | −30,935, −20,424, −5661, −363 | −363 | |
| *Stat2* | −172 | −172 | |
| *Zfp608* | −219,854, −167,598, −146,281, −144,047, −38,299, −2925, −1026, +18,808, +19,198, +32,325, +78,559, +83,469, +87,457, +110,795, +143,065, +145,788, +161,945, +487,266 | −1026 | +18,808 |

The last two columns show the closest upstream (−) and downstream (+) peak locations compared to the TSS.

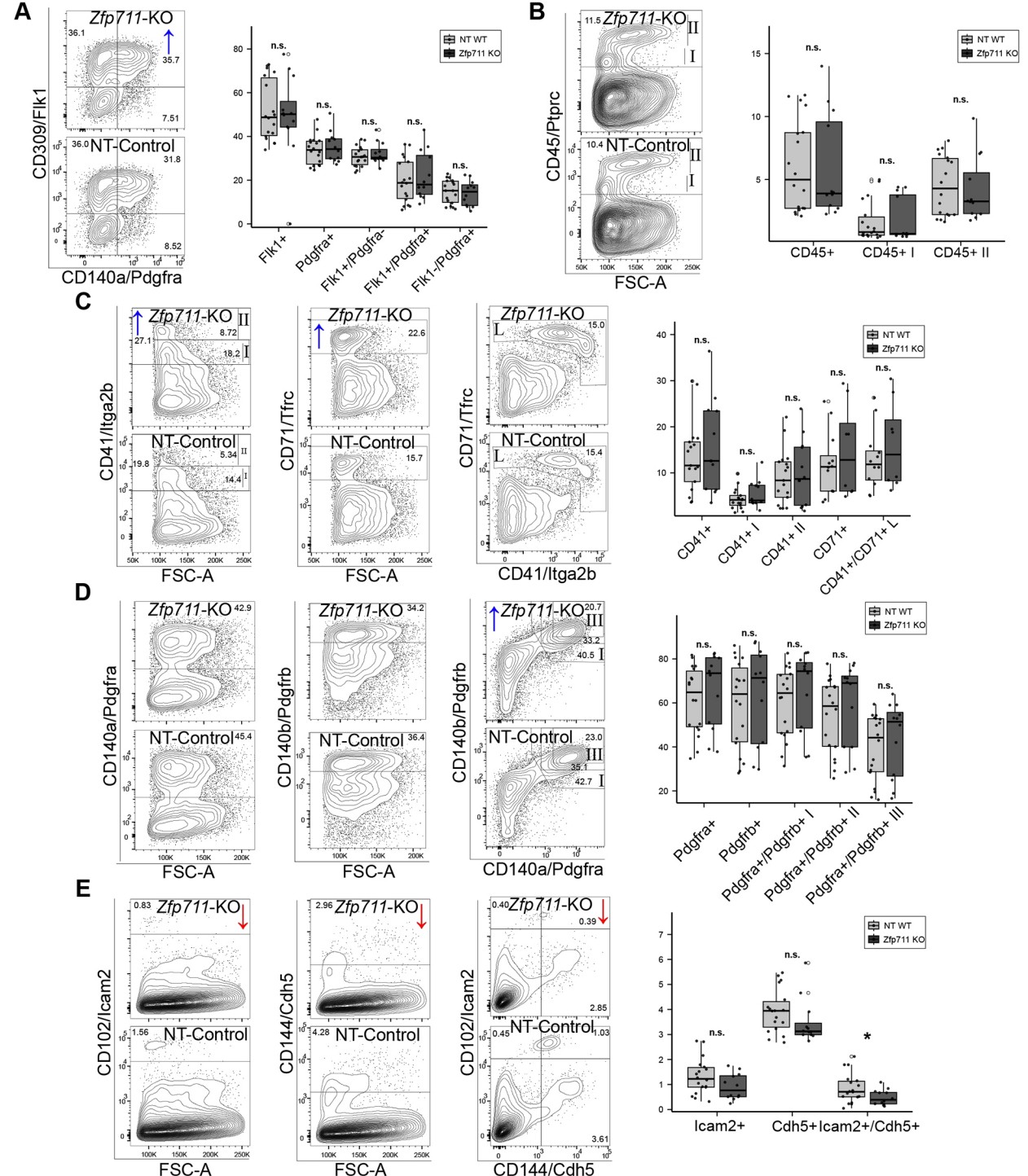

**Fig. 6. Flow cytometric validation of *Zfp711* knockout effects on mesodermal and hemato-endothelial differentiation.** (A) D4 analysis showing distribution of hemato-endothelial progenitors (Flk1$^+$/Pdgfra$^-$) and mesodermal populations (Flk1$^+$/Pdgfra$^+$) in Δ*Zfp711* versus NT-Control cells. Representative plots (left) and quantification (right) demonstrate increased Fkl1$^+$/Pdgfra$^+$ populations. (B) D7 EMP analysis using CD45 marker. (C) D7 erythroid lineage assessment using CD41 and CD71 markers. (D) D7 late mesoderm populations marked by Pdgfra and/or Pdgfrb. (E) D7 analysis of endothelial markers. Representative plots and quantification show reduction in Cdh5$^+$/Icam2$^+$ endothelial cells in Δ*Zfp711*. Boxes indicate the interquartile range (IQR; 25th-75th percentile), the central line indicates the median, whiskers extend to 1.5× IQR, and points beyond represent outliers. Data are from three independent experiments with four knockout and six control clones. *$P$<0.05; n.s., not significant.

Paradoxically, despite *Atf3* downregulation, interferon pathways were suppressed rather than activated, as in Δ*Atf3* cells. This divergence results from differential regulation of interferon effectors (*Bst2* and *Ifitm3*) that are upregulated in Δ*Atf3* cells but downregulated in Δ*Zfp711* cells, demonstrating that Zfp711 has Atf3-independent functions in immune regulation.

In summary, loss of Zfp711 caused a striking shift in mesodermal populations: the limb mesoderm-allantois (Hoxa$^+$)

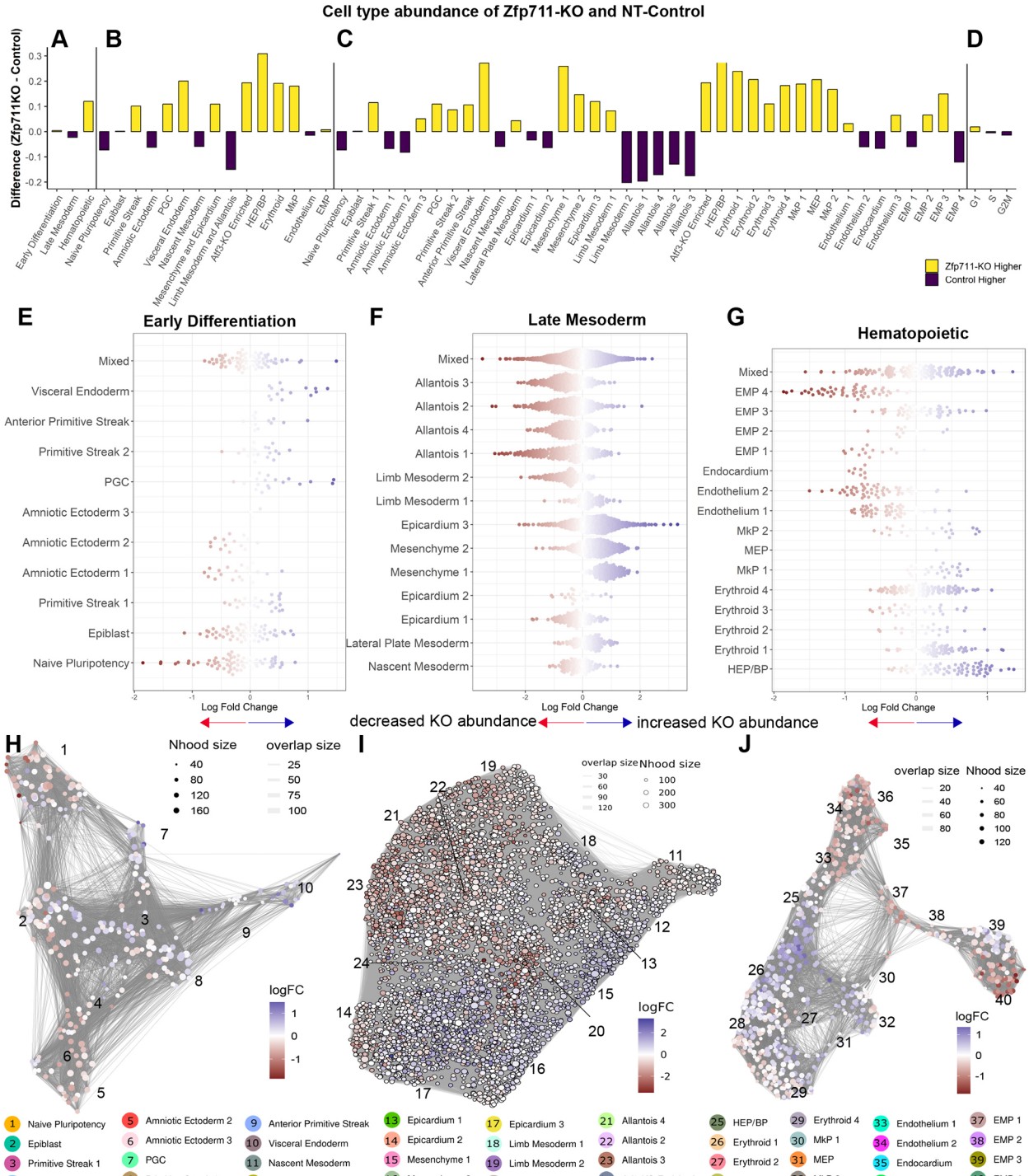

**Fig. 7. Single-cell differential abundance analysis reveals *Zfp711*-dependent changes in cell population dynamics.** (A-D) Cell type abundance differences between Δ*Zfp711* and NT-Control across hierarchical annotations. Yellow bars indicate higher abundance in Δ*Zfp711*; purple bars indicate lower abundance. (A) Group level, (B) cluster level, (C) sub-cluster level, (D) cell cycle phases. Proportions of the cell types produced with Speckle/Propeller. (E-G) MiloR k-nearest neighbor analysis showing log fold-changes in cell abundance by group with subcluster annotation. (E) Early differentiation populations, (F) late mesoderm (G) hematopoietic lineages (Spatial FDR<0.1). (H-J) Neighborhood graph visualization of differential abundance produced with MiloR. (H-J) Cluster-level network showing relationships between Early differentiation, late mesoderm and hematopoietic populations. UMAP with neighborhood overlay (node size=neighborhood size). Color scale indicates log fold-change (red, decreased; blue, increased in Δ*Zfp711*).

population decreased, while the mesenchyme-epicardium limb mesoderm-allantois (Hoxb[+]) increased. Hematopoietic differentiation was enhanced, with blood progenitors and erythroid cells increasing, while endothelial cells decreased.

Notably, *Atf3* was downregulated in Δ*Zfp711* cells. Unlike Δ*Atf3* cells, interferon pathways were suppressed rather than activated, indicating *Zfp711* acts upstream of *Atf3* but also has independent functions.

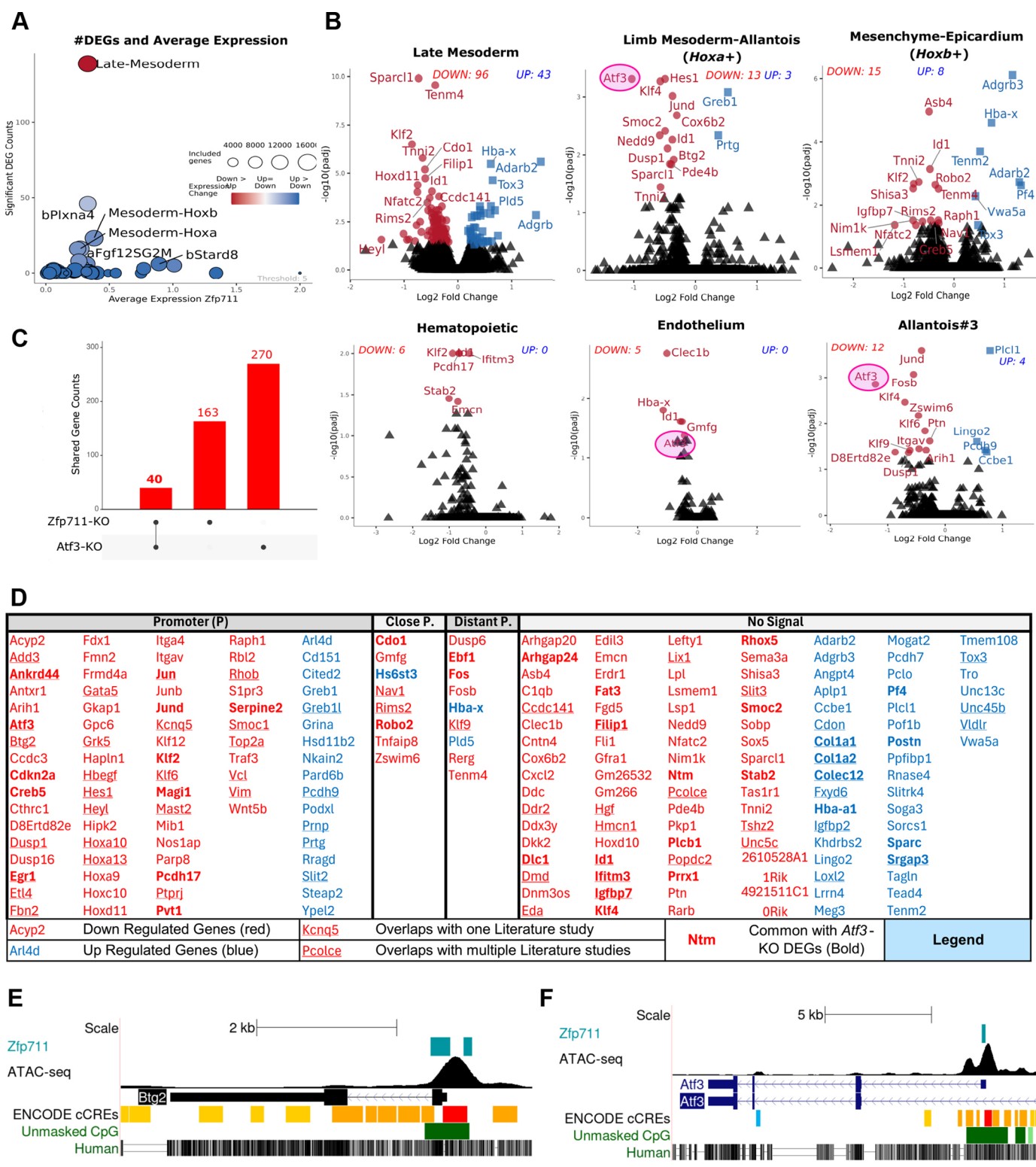

**Fig. 8. Results and categorization of the differential gene expression analysis of the ΔZfp711 cells.** (A) Differential gene expression [(adjusted-*P*<0.05)] across cell type hierarchies. Circle size indicates genes analyzed; color shows upregulation (blue) or downregulation (red). Late mesoderm shows the highest DEG count (139 genes). (B) Volcano plots revealing distinct transcriptional changes: *Hoxa*+ mesoderm (limb mesoderm-allantois) shows downregulation of *Atf3* (*Hes1*, *Klf4*, *Dusp1* and *Jund*), while *Hoxb*+ mesoderm (mesenchyme/epicardium) shows different gene signatures with *Tox3* and *Adgrb3* upregulation. Endothelium and Allantois#3 cell populations also shows the *Atf3* downregulation. (C) Overlap analysis identifying 40 shared DEGs between ΔZfp711 and ΔAtf3. (D) Comprehensive DEG categorization based on *Zfp711* binding proximity. Promoter-bound targets include *Atf3*, *Jun* and *Jund*, suggesting direct regulation. A single underlined gene shows that the gene is shared with DEGs from other studies in the literature once. A double underlined gene shows that the gene is shared with DEGs from other studies in the literature more than once. (E,F) Genomic tracks showing *Zfp711*-binding sites downstream of TSS with corresponding ATAC-seq peaks. (E) *Zfp711* locus. (F) *Atf3* promoter showing binding, which may explain *Atf3* downregulation in ΔZfp711 cells.

**Table 3. ΔZfp711 cells: selected DEGs (TFs and other important genes)**

| | |
|---|---|
| Late mesoderm | *Klf2, Id1, Nfatc2, Creb5, Btg2, Tead4, Prrx1, Jund, Gata5, Sox5, Dusp1, Rarb, Fli1, Rhox5, Heyl, Hoxa9, Hoxa10, Hoxa13, Hoxd10, Klf4, Klf6, Klf12, Id3,* **Foxp1, Crtc3** and **Hoxa11** |
| *Hoxa*⁺ mesoderm | *Atf3, Hes1, Klf4, Jund, Id1, Dusp1, Btg2,* **Klf6, Sox5, Nfatc2, Hoxb8** and **Creb5** |
| *Hoxb*⁺ mesoderm | *Id1, Klf2, Creb5, Nfatc2, Tox3, Id3, Zfpm1, Nfib, Tead4, Btg2* and *Hipk2* |
| Epicardium#3 | *Nfatc2, Hipk2, Tshz2, Hoxc10, Klf6, Nfib, Rarb, Rhox5* and *Meis2* |
| Allantois#3 | *Jund, Fosb, Klf4, Klf6, Klf9, Dusp1,* **Hes1** and **Hoxd11** |
| Hematopoietic | *Id1* and *Klf2* |
| Endothelium | *Atf3, Id1, Hes1, Ier2* and *Jun* |
| Also in Δ*Atf3* | *Atf3, Ebf1, Egr1, Fos, Id1, Jun, Jund, Klf2, Klf4, Prrx1, Rhox5* and **Creb5** |

Adjusted-*P* values are between 0.05 and 0.1 for the genes in bold, and <0.05 for the genes in italics.

### BCL6B

#### Δ*Bcl6b* cells show no observable changes

*Bcl6b* is expressed in endothelium (Figs S30-S32). It was inactivated as shown in Fig. S28A. Δ*Bcl6b* cells showed minimal changes compared to controls in terms of cell composition, as determined by scRNA-seq (Figs S33A-G, S34, S35; Tables S2Q-X, S3G-I). Surface marker staining (Fig. S36; Table S1E,F) did not change significantly. The DGEA of groups, clusters and sub-clusters, respectively, showed only 18 genes affected by the KO of *Bcl6b* (Fig. S33H,I; Table S4U,V). These KO cells do not appear to display any statistically significant change in the tissue where *Bcl6b* is expressed highly, i.e. only a small decrease was seen in endothelium (Fig. S33B,C,G).

### DISCUSSION

#### Technical considerations

Our multiplexed scRNA-seq approach successfully captured 71,107 cells across continuous differentiation with minimal batch effects, enabling parallel analysis of three TF knockouts with biological replicates. However, several technical limitations warrant consideration. Naive pluripotent ESCs showed reduced CMO labeling efficiency, while erythroid cells and EMPs exhibited higher doublet rates, possibly due to membrane changes affecting cell-cell interactions (Fig. S37). The extended labeling procedure may have compromised detection of certain cell types; cardiac lineages that were identified in our previous non-multiplexed D7 scRNA analysis were absent in this dataset, despite EB pulsation indication that cardiomyocytes are present.

Quality control of scRNA-seq data is important, as certain cell types may get lost. For example, use of the ratio of 'detected gene number' (nFeature-RNA) to 'unique RNA-molecule count' (UMI, nCount-RNA) can lead to discarding highly specialized cells, when present in low numbers. We did not use this ratio for preprocessing (filtering low RNA and gene counts, doublet removal, etc.), because erythroid lineages showed a low 'detected gene number' to 'unique RNA-molecule count' ratio because erythroid cells have a large bias of globin mRNAs. Such cells are filtered out when this ratio is used in the quality control of single cells.

DGEA frequently returns top-upregulated genes such as *Fam71a* (Δ*Atf3*), *Pof1b* (Δ*Zfp711*) and *Slc16a13* (Δ*Bcl6b*). These three genes may be affected by the deletion of part of the neighboring DNA, causing changes in the local transcription dynamics (e.g. read-through, deletion of a regulatory region, reduced distances between promoters and upstream enhancers).

#### *Atf3* functions as a dual regulator of mesodermal and hematopoietic development

Loss of *Atf3* revealed two distinct phenotypes: enhanced mesodermal differentiation (10-28% increase) and impaired EMP formation (22% decrease). The EMP defect results from delayed EHT and downregulation of crucial hematopoietic regulatory genes (*Runx1*, *Mafb* and *Sox4*), with *Mafb* containing multiple direct Atf3-binding sites. Our findings align with those of Yin et al. (2020), who reported

**Table 4. ΔZfp711 cells: selected DEGs and neighboring Zfp711 peaks with GREAT single nearest gene assignment**

| Gene | Single nearest gene assigned *Zfp711* peaks | Closest upstream | Closest downstream |
|---|---|---|---|
| *Atf3* | −41,032, +104 | −41,032 | +104 |
| *Btg2* | −299, +94, +18,070, +18,425 | −299 | +94 |
| *Cited2* | −777, −47, +300 | −47 | +300 |
| *Creb5* | +186 | | +186 |
| *Egr1* | +34, +27,443, +27,827 | | +34 |
| *Gata5* | −151, +19 | −151 | +19 |
| *Hbegf* | −305, −160, +85, +455 | −160 | +85 |
| *Hes1* | −119,243, −118,817, +17, +1052, +5161 | −118,817 | +17 |
| *Heyl* | +51 | | +51 |
| *Hoxa10* | +808 | | +808 |
| *Hoxa13* | −272, +570, +966 | −272 | +570 |
| *Hoxa9* | −3196, −2299, +1292 | −2299 | +1292 |
| *Hoxc10* | +68 | | +68 |
| *Hoxd11* | −103,470, −26,354, −26,154, −14,319, −13,794, −9459, −2524, +61, +1402 | | +61 |
| *Jun* | −226,412, −109,275, −99,651, −24,349, −987, −514, −310, +64 | −310 | +64 |
| *Junb* | +27 | | +27 |
| *Jund* | −241, +211 | −241 | +211 |
| *Klf2* | −258, +194 | −258 | +194 |
| *Klf6* | −336, +110 | −336 | +110 |
| *Pard6b* | −601, +300, +34,150, +36,345, +36,608, +47,529, +47,817 | −601 | +300 |

The last two columns show the closest upstream (−) and downstream (+) peak locations compared to the TSS.

decreased EMPs and cardiac abnormalities, though we could not assess specific cardiac lineages due to technical limitations.

The emergence of interferon-responsive populations (ΔAtf3-enriched mesoderm and endothelium#3) suggests that Atf3 normally suppresses interferon signaling during development. GSEA suggests that the top-hallmark results are 'Cell cycle and related pathways', which are upregulated in ΔAtf3 EMP, but this is not seen in 'late mesoderm'. The 'interferon response' is the top-hallmark,

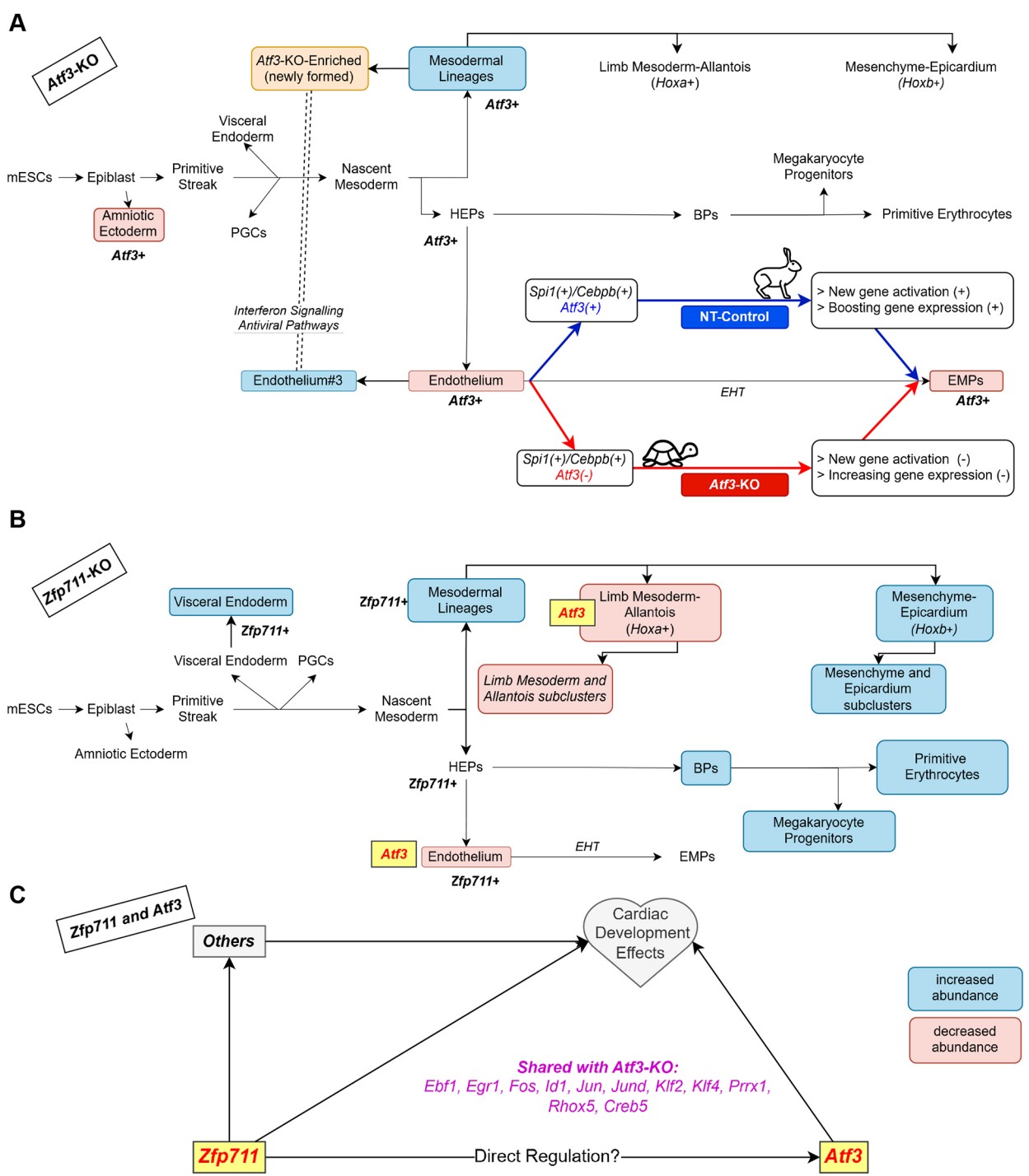

Fig. 9. Summary of *Atf3* and *Zfp711* knockout effects on mESC differentiation in the *in vitro* hemato-endothelial differentiation. (A) Summary of ΔAtf3 effects. Loss of *Atf3* causes: (1) emergence of interferon-responsive ΔAtf3-enriched mesoderm, (2) reduced amniotic ectoderm, and (3) impaired EHT with decreased endothelium and EMPs (turtle symbol indicates delayed transition). The lower pathway shows a mechanistic model where *Atf3* cooperates with *Spi1* and/or *Cebpb* to drive EMP formation. Hare symbol indicates normal rapid differentiation. (B) Summary of ΔZfp711 effects. Loss of *Zfp711* causes: (1) decreased Hoxa+ mesoderm (limb mesoderm-allantois), (2) increased Hoxb+ mesoderm (mesenchyme/epicardium), (3) enhanced erythropoiesis and megakaryopoiesis, and (4) decreased endothelium. *Atf3* downregulation in Hoxa+ mesoderm and endothelium suggests hierarchical regulation. (C) Potential interactions between *Zfp711* and *Atf3*. Both genes are implicated in regulating gene expression related to cardiac development. *Zfp711* might directly regulate *Atf3*, and both genes share a set of target genes, e.g. *Ebf1*, *Egr1* and *Fos*.

while 'cell cycle gene sets' is not in the top 20. The likely explanation for these opposing results is that the ΔAtf3 has much fewer cells in the 'EMP#4' sub-cluster. The total number of mutant cells in the G1 phase will therefore also be lower than in the NT-Control.

In late mesoderm, predominantly upregulated DEGs indicate that Atf3 functions as a transcriptional repressor, particularly of TGFβ and/or BMP (*Id1*, *Smad7* and *Id2*) and TNFα-NFκB pathways (*Id2*, *Plk2*, *Klf4*, *Fos* and *Jun*). The upregulation of interferon signaling and collagen formation aligns with previous reports (Sood et al., 2017); positive enrichment was clear for the 'signal transduction' category, i.e. in Hedgehog family signaling, in the Erad pathway, and in *Runx2* expression and activity pathways. In contrast, negative enrichment was observed in the MAPK and Erk1-Erk2 cascade, in TNF signaling, in serine/threonine-kinase signaling, in TGFβ, in BMP, in downstream SMADs, in apoptotic signaling, and in P53-related pathways, which explains why the 'development' category is downregulated.

Our chromatin accessibility analysis provides mechanistic insight into how *Atf3* regulates the EHT (Fig. 9A). The emergence of novel enhancer peaks during EMP formation suggests dynamic regulatory rewiring at this crucial developmental transition. Motif analysis revealing Spi1-binding sites adjacent to Atf3 sites indicates these factors work cooperatively.

*Spi1* and *Cebpb* are induced specifically during EHT, when *Atf3* expression also peaks. This coordinated upregulation suggests a feedforward regulatory circuit that reinforces hematopoietic commitment. Such cooperative binding between ubiquitously expressed factors like *Atf3* and lineage-specific factors like *Cebpb* represents a common developmental strategy to achieve cell-type specificity.

The delayed EHT observed in ΔAtf3 cells can thus be explained by disruption of this cooperative network. Without *Atf3*, newly expressed hematopoietic factors cannot efficiently activate their target genes, resulting in accumulation of immature EMPs and reduced mature populations. This model aligns with established roles for Spi1/PU.1 as a master regulator of hematopoiesis (Iwasaki et al., 2005) and extends our understanding of how stress-responsive factors like Atf3 integrate with developmental programs.

### *Zfp711* and mesodermal patterning through *Atf3*
*Zfp711* deletion primarily affected late mesoderm, causing a striking shift from limb mesoderm-allantois (*Hoxa*$^+$) to mesenchyme-epicardium (*Hoxb*$^+$) populations (Fig. 9B). Mesoderm-allantois showed downregulation of multiple TF genes, including *Atf3*, *Klf4*, *Klf6*, *Klf9* and *Hes1*, and growth factors (*Dusp1* and *Hbegf*). The identification of Zfp711-binding sites in the *Atf3* promoter, combined with 40 shared DEGs between the two knockouts, supports a *Zfp711→Atf3* regulatory hierarchy consistent with Rhie et al. (2018) (Fig. 9C).

Our findings complement those of Snabel et al. (2025) who reported cardiac differentiation defects in ZNF711-knockout human PSCs. We identified 37 shared DEGs, including cardiac-essential genes (*Dmd* and *Unc45b*), and changes in retinoic acid receptor genes (*Rarg* and *Rarb*), supporting a role for *Zfp711* in cardiac development. However, unlike ΔAtf3, blood lineages increased in ΔZfp711 because the expression of key hematopoietic genes (*Runx1* and *Mecom*) remained intact, demonstrating *Zfp711* has both *Atf3*-dependent and -independent functions.

The endocardium sub-cluster showed distinct marker expression (*Gata4*, *Hand2*, *Tbx20* and *Sox6*) with *Zfp711* downregulation, suggesting *Zfp711* regulates endothelial-to-endocardial transitions (Daniel et al., 2020; Pawlikowski et al., 2019).

### *Bcl6b* shows minimal impact during early development
*Bcl6b* deletion produced only 18 DEGs with no significant phenotype and effectively serves as an internal control that validates our multiplexing approach. *Bcl6b* appears to have almost no role in early differentiation, although its absence was shown to affect neovascularization in ocular vascular diseases (Tanaka et al., 2023), a tissue that is absent in the mESC system.

### Conclusions and limitations
Our *in vitro* ESC differentiation system successfully recapitulates major features of hemato-endothelial development, but it cannot fully replicate the complex *in vivo* microenvironment. The broad developmental effects observed for *Atf3* and *Zfp711* would need tissue-specific conditional knockouts to definitively establish their functions during embryonic hematopoiesis. Furthermore, our scRNA-seq captures only D4 and D7 timepoints representing peak hemato-endothelial specification; examining additional timepoints with ChIP-seq and ATAC-seq profiling would definitively establish the dynamic regulatory hierarchies and sequential binding events that govern these TF networks and establish whether a *Zfp711* to *Atf3* hierarchy is suggested by our data. Such approaches would provide a more-complete mechanistic understanding of how these factors coordinate hemato-endothelial specification.

Nevertheless, our study demonstrates that multiplexed scRNA-seq with orthogonal validation can efficiently dissect TF functions during differentiation. Our integrated analysis of ATAC-seq, ChIP-seq and expression data addressed three key items.

(1) Early lineage decisions: *Atf3* promotes hemato-endothelial specification while restraining mesoderm; *Zfp711* patterns mesodermal subtypes; *Bcl6b* has no observable role.

(2) Later differentiation: *Atf3* is essential for EMP maturation through EHT; *Zfp711* affects endothelial and cardiac lineages partially through *Atf3* regulation.

(3) Molecular mechanisms: *Atf3* directly regulates hematopoietic transcription factors while suppressing interferon responses; *Zfp711* appears to act upstream of *Atf3* but maintains independent functions in immune regulation.

(4) Novel technical approach: multiplexed scRNA-seq with flow cytometry provides a cost-effective, batch-free analysis of multiple TFs by *in vitro* differentiation.

## MATERIALS AND METHODS
### Cell line generation with CRISPR-Cas9 knockouts
Two guide RNAs were designed to delete each of the three genes using the IDT-Custom Alt-R CRISPR-Cas9 guide RNA tool, including two non-targeting gRNAs. The 'Alt-R CRISPR-Cas9 System: Cationic lipid delivery of CRISPR ribonucleoprotein complexes into mammalian cells' guidelines of IDT [version 6 (v6)] followed downstream analysis with Alt-R CRISPR-Cas9 tracrRNA. ATTO 550 (1075927), Alt-R CRISPR-Cas9 crRNA, Alt-R S.p. Cas9 Nuclease V3, (1081058) and 1075927 Lipofectamine CRISPRMAX Cas9 Transfection Reagent (CMAX00008) were used for the transfection of mESCs. Transfected cells were sorted after 24 h into 96-well plates, which were MEF coated and contained 2i+mESCs Medium. After 5-7 days, each well was checked for monoclonal growth, transferred into a new 96-well plate and two copies of a 96-well plate were prepared; one plate was frozen down within 10% DMSO in FBS, and the other plate was used for genomic DNA isolation. Following PCR genotyping, KO-positive clones were thawed, expanded, genotyped again and frozen in multiple vials.

Guide RNAs used to target were as follows: *Atf3*, GGGACTAGTTTT-CACAACGCTGG and TCATTATTGAGGTTGTCCAATGG; *Bcl6b*, AGGTACCAGACTTTACCTGGGGG and AAGGCCCTGGTACCAAC-TACAGG; *Zfp711*, GATGGGATAACTCTCGATCATGG and GTTAA-GACACCTTATTTGATTGG; NT-Control, IDT-control crRNAs (1072544, 1072545).

The primers for PCR3 genotyping of *Atf3*, *Zfp711* and *Bcl6b* were as follows. For *Atf3*, the primers were Atf3_out_f5_#601 (CTGGAGA-CCTCTGTGCAAGA), Atf3_out_r3_#602 (TGCCCTGTCACTGAGTA-TGG), Atf3_ins_f2_#604 (ACAGCTTGCCATGAAACCTG) and Atf3_ins_r1_#605 (CTCCTCAATCTGGGCCTTCA). For *Zfp711*, the primers were Zfp711_out_f4_#577 (CCTGGTTTTGGCCTTACAGA), Zfp711_out_r2_#578 (TGCATTTTCTTTCCCCACCC), Zfp711_ins_f2_#580 (AAACTGCCGAACAAGGACTG) and Zfp711_ins_r2_#581 (TCACAA-TGCCTACACTGGTG). For *Bcl6b*, the primers were Bcl6b_out_f3_#583 (CTCATCCTCGGGTGCTTAT), Bcl6b_out_r3_#584 (TCAAACCCAG-CTGTTCATGC), Bcl6b_ins_f2_#586 (GGCAGTTCTTATCGCTTGCA) and Bcl6b_ins_r1_#587 (ACACCGTCTCCTAGTCGTTG).

## Media

mESC medium consisted of DMEM HyClone (Cytiva, SH30081), 15% fetal bovine serum (FBS) (Capricorn Scientific FBS-12A, CP18-2152), 1×MEM Non-Essential Amino Acids Solution (100×) (Gibco 11140-035), 2 mM - GlutaMAX Supplement (Gibco 35050061), 10 mM HyClone HEPES Buffer (Cytiva, SH30237.01), 1×penicillin-streptomycin (100×) (Sigma-Aldrich P0781), 0.1 mM 2-mercaptoethanol (50 mM) (Gibco, 31350010) and 1000 U.ml ESGRO Recombinant Mouse LIF Protein $10^7$ units/ml (Sigma-Aldrich, ESG1107). MEF medium consisted of DMEM (Capricorn Scientific DMEM-HPSTA), 15% FBS (Capricorn Scientific FBS-12A, CP18-2152), penicillin-streptomycin (100×) (Sigma-Aldrich P0781) and 2 mM GlutaMAX Supplement (Gibco 35050061). 2i+mESCs medium consisted of mESCs-Medium with 3 µM - CHIR 99021 (biotechne-TOCRIS 4423) and 1 µM PD0325901 (selleckchem S1036).

Freezing medium consisted of FBS (Capricorn Scientific FBS-12A, CP18-2152) with 10% DMSO (general lab consumable). Dissociation buffers consisted of Trypsin-EDTA (0.05%) and phenol red (Gibco, 25300054), or TrypLE Express Enzyme (1×) and phenol red (Gibco, 12605010). Differentiation medium consisted of IMDM, GlutaMAX Supplement (Gibco, 31980048), 15% FBS (Capricorn Scientific FBS-12A, CP18-2152), 1×penicillin-streptomycin (100×) (Sigma-Aldrich, P0781), 0.1 mM 2-mercaptoethanol (50 mM) (Gibco, 31350010), 50 µg/ml L-ascorbic acid (Sigma-Aldrich, A5960-25G) and 150 µg/ml transferrin (Roche, 10652202001). Wash buffer consisted of 10% FBS (Capricorn Scientific FBS-12A, CP18-2152) in PBS (Capricorn Scientific PBS-1A or Gibco 14190-094).

## mESCs and *in vitro* differentiation

129X1/Svj mESCs (Jackson Laboratory) were grown on irradiated MEFs at 37°C and 5% $CO_2$ and split every 2 days. Two or three passages were carried out before differentiation: 2i+mESCs medium was used for one passage and then switched back to mESCs medium. At starting the differentiation, the mESCs were harvested and made into a single-cell suspension, strained with a 40 µM cell strainer (Greiner 542040), washed three times and counted in Trypan Blue (Bio-Rad 1450021). One million live cells were plated in 12 ml freshly prepared differentiation medium and plates (Greiner 633102) were placed on an orbital shaker at 60 rpm at 37°C under 5% $CO_2$. On day 2 of differentiation, 2 ml of prewarmed differentiation medium was added to each plate. On D4, HEP and mesodermal cell differentiation efficiency was checked using half of the embryonic bodies. The remaining half was transferred to new plates, 13 ml differentiation medium was added and the conditions maintained until D7.

The harvested EBs were washed three times with PBS, and prewarmed TrypLE was added in a 50 ml falcon tube and placed in a 37°C water bath while shaking with intermittent pipetting for dissociation. 3 ml Trypsin/EDTA was added after 4 min to dissociate the EBs for at least 7 min, checking the dissociation visually. The disassociation was inactivated with PBS+10%FCS, strained with a 40 µM cell strainer to obtain a single-cell suspension, washed three times and stained for cell surface markers or prepared for scRNA-seq.

## Surface marker staining and flow cytometric analysis

On D4, Flk1 and Pdgfra antibodies were used, while on D7, CD144, CD102, Pdgfra, Pdgfrb, CD41, CD71, CD45 and CD93 antibodies were used. Briefly, 1 million cells were incubated in 100 µl PBS+10%FCS with the

antibody combinations for 30 min on ice, followed by three washing steps. Stained samples were measured with a BD LSRFortessa Cell Analyzer. The antibodies were used following the manufacturer's recommendation: CD309 APC (BioLegend, 136405), CD309 AF647 (Biolegend, 121910), CD140a PE (Biolegend, 135905), CD140a PE/Cy7 (Biolegend, 135911), CD140b APC (Biolegend, 136007), CD102 AF647 (Biolegend, 105611), CD144 PE/Cy7 (Biolegend, 138015), CD41 PE (Biolegend, 133905), CD93 PE (Biolegend, 136503), CD45 APC (Biolegend, 157605), CD45 APC/Cy7 (Biolegend, 103115), CD71 APC (Biolegend, 113820) and CD202 PE (Biolegend, 124007).

Flow-cytometric analysis results were analyzed using FlowJo (v10.8.1 and v10.10.0) software. Staining panel frequencies were compared to mixed-model statistical cells using the lme4 R package (v1.1-35.5) (Bates et al., 2015) with the setting "~ Conditions+(1|Differentiation)".

## Differentiation for scRNA-seq and CMO multiplexing

For scRNA-seq, three clones were selected for each condition (ΔAtf3, ΔBcl6b, ΔZfp711 and NT-Control), and three replicate differentiation plates were generated for each clone. The plates were merged on D4, and the differentiation continued until D7. After harvesting EBs and generating a single-cell suspension, samples were washed three times and the cells were counted. The "3′ CellPlex Kit Set A" (10X Genomics, 1000261) was used according to the manufacturer's instructions to label the samples. The cell concentration was measured again, and samples were pooled, targeting equal cell numbers to proceed to the single-cell RNA seq capture.

## scRNA-seq capture, library preparation and raw data analysis

scRNA-seq was performed using the 10X Genomics Chromium X Platform. The Chromium Next GEM Single Cell 3'Reagent kit v3.1(Dual indexing) kit with feature Barcode technology for cell multiplexing (10X Genomics) was used following the manufacturer's instructions, generating separate gene expression and cell multiplexing libraries. These libraries were subsequently sequenced on a Novaseq 6000 platform (Illumina) with cycles setting 28-10-10-90 cycles. Approximately, 25,000 reads/cell and 5000 reads/cell were generated for, respectively, each of the GEX and CMO libraries. The raw data were demultiplexed using the CellRanger-7.0.1 mkfastq pipeline. The gene expression data were processed using the CellRanger-7.0.1 count pipeline. Gene expression data in combination with CMOs were processed using the CellRanger-7.0.1 multi pipeline. The mm10 reference genome was used for generating the counts matrix.

## scRNA-seq analysis

scRNA-seq data were analyzed using the Seurat v5 R package (v5.0.1, v5.3.0) (Butler et al., 2018; Hao et al., 2021, 2024; Satija et al., 2015; Stuart et al., 2019). In addition to the CellRanger-7.0.1 multipipe line's demultiplexing, a second demultiplexing step was performed using the HTODemux function with a positive quantile=0.99 threshold. Next, Median Absolute Deviation thresholding (Germain et al., 2020; Heumos et al., 2023) was used to remove low-quality cells with nFeature_RNA (the number of detected genes), nCount_RNA (the amount of detected RNA), and percent.mt (percentage of mitochondrial RNA) with thresholds 5 (initial threshold) and 3 (final threshold). Ambient RNA, extracellular free-floating RNA that causes background counts, was removed with SoupX (Young and Behjati, 2020) using default parameters. The scDblFinder package (Germain et al., 2021) was used to detect doublets, and 0.20 was set as the doublet threshold. Next, the five libraries were merged and combined into one dataset using the JoinLayers function.

Temporary mapping and clustering of the merged dataset was performed before removing low-quality cells with the following functions of the Seurat: NormalizeData, ScaleData, FindVariableFeatures (vst, nFeatures=4000), RunPCA, FindNeighbors (dims=1:75), RunUMAP (dims=1:75, spread=2), FindClusters(resolution=4, algorithm=4, n.start=100, n.iter=100) and CellCycleScoring. Low-quality cells were removed using the following classifications: singlet-assigned cells from the Cell Ranger pipeline's "Confidence Assignment Table"; singlet-assigned cells from HTODemux; singlets assigned cells from scDblFinder and a Median Absolute Deviation threshold of 3. Following this, cell numbers from clusters were compared before and after removing the low-quality cells, cell clusters containing mostly

low-quality cells were also removed from the dataset. Markers of the cell types were detected using the FindMarkers/FindAllMarkers functions of Seurat.

Next the following steps were performed: cell cycle regression with ScaleData, FindVariableFeatures (nFeatures=4000), RunPCA, RunUMAP(dims=1:75, spread=1, n.neighbors=50, n.epoch=1000), FindNeighbors(dims=1:75, k.param=50), FindClusters(resolution=1.25, algorithm=4, n.start=100, n.iter=100). Clusters that require higher resolution were further divided using the FindSub-clusters function. Later, three levels of cell annotation were generated: groups, clusters and sub-clusters. The groups were then divided into three categories: "Early differentiation", "Late Mesoderm" and "Hematopoietic". Each of the three going through FindVariableFeatures, RunPCA(npcs=50), FindNeighbors(dims=1:50) and RunUMAP(dims=1:50). Markers of the annotated cells were detected with FindMarkers and FindAllMarkers function.

For the DGEA, each TF-KO was compared to the NT-Control with pseudo-bulked DESeq2 (v1.42.1) (Love et al., 2014) in all cell type annotations in a three-level resolution (groups, clusters and sub-clusters). Single-cell expression counts were converted to a pseudo-bulk counts matrix with Seurat's "AggregateExpression" function. Next, a DESeq2 object was created using the "DESeqDataSetFromMatrix", a function of DESeq2. Genes with low read counts were removed. The "DESeq" function is applied without extra arguments, and adjusted-$P<0.05$ is used to detect the differentially expressed genes (DEGs).

The marker detection of the three small sub-clusters (Endothelium#3, Endocardium and $\Delta Atf3$-Enriched) performed with Seurat's FindMarkers function was utilized with the Wilcox method with min. pct and log.fc.threshold equal to 0. Gene set enrichment analysis (GSEA) was performed with the fgsea R package (v1.29.2) (Korotkevich et al., 2021 preprint) on MsigDB's (v2023.2) (Mootha et al., 2003; Subramanian et al., 2005) Hallmark, GO-Biological Processes and Reactome gene sets. Results of the DGEA with "stat" values were used to rank the gene list. Low-expression genes were removed if a gene had a base value of less than 15. The fgsea function was run with minSize=15 and maxSize=500 settings. The top 20 results of each gene set were used for visualization. Results that showed adjusted-$P<0.001$ were manually curated into categories (i.e. metabolism and signal).

Speckle (Phipson et al., 2022) and miloR (Dann et al., 2022) packages were used for DAA (or Compositional Analysis). For Speckle, default settings were used with the propeller function.

For MiloR, the following process was repeated for each KO-TF compared to Control in EarlyDiff, Late-Mesoderm and Hematopoietic groups. KNN graphs were constructed with "buildGraph" (parameters: k=35, d=50) and "makeNhoods" (prop=0.3, k=35, d=50, refined=TRUE). The "k" value was calculated with the ≥5×(number of samples) formula. Euclidian distances were calculated with "calcNhoodDistance" (parameters: d=50). Differential neighborhood abundance was tested with "testNhoods". The dependency between the average number of cells per sample and the logFC was checked using the "plotNhoodMA" function with MA-Plot. With "buildNhoodGraph", an abstract graph of the neighborhoods visualization was created and visualized with "plotNhoodGraphDA" (parameters= alpha (SpatialFDR)=0.1). The "annotateNhoods" function was used to annotate the neighborhoods with the sub-clusters. In cases when the neighborhoods did not contain at least 50% of the majority from any sub-clusters, they are labeled as "Mixed". "plotDAbeeswarm" was used to visualize the distribution of log fold changes across neighborhood annotations. "findNhoodGroupMarkers" was used for marker gene detection on selected neighborhoods (SpatialFDR <0.1).

The in vivo scRNA-seq mouse gastrulation atlas (Pijuan-Sala et al., 2019) and extended atlas (Imaz-Rosshandler et al., 2023) were used for visualization, aligning in vitro and in vivo data through the FeaturePlot function of Seurat. Pseudotime construction was performed with Palantir (Setty et al., 2019) and Slingshot (Street et al., 2018).

Cell type mapping and annotation: cell types were annotated by mapping to the published reference dataset (Imaz-Rosshandler et al., 2023) through the following preprocessing steps: genes were limited to those present in both query and reference datasets; uninformative genes were removed (mitochondrial, ribosomal, hemoglobin, imprinted genes, Y chromosome genes, Xist and Tsix); low-expressed genes were removed from both datasets; and reference and query datasets were limited to remaining genes. Seurat's FindTransferAnchors and TransferData functions were used for mapping cell types. Leiden clusters were assigned labels based on the most frequently mapped cell type.

Pseudotime and trajectory analysis: developmental trajectories and pseudotime were analyzed using Slingshot (Street et al., 2018) for both in vivo and in vitro datasets. Lineage-specific subsets were extracted from the complete dataset and new UMAP embeddings were generated for each lineage subset. Trajectories were defined using getLineages (dist.method='slingshot') with specified start and end clusters, and curves were fitted using getCurves function. For computational efficiency, large clusters were downsampled to ~2000 cells per cell type. Expression of selected TFs (Atf3, Zfp711 and Bcl6b) was plotted along each lineage trajectory.

EMP motif analysis: novel EMP enhancer peaks were identified by manual inspection of chromatin accessibility at top upregulated EMP genes. Genomic coordinates of selected regions were extracted and motif scanning was performed using FIMO (Grant et al., 2011) with the JASPAR2024 CORE database (Rauluseviciute et al., 2024) (parameters: reverse strand search enabled, P-value cutoff=0.0001). For the publicly available Atf3, Spi1 and Cebpb DNA binding data, ReMap2022 was used (Hammal et al., 2022).

## DEG, literature and ChIP-seq integration

Public human and mouse datasets were used to compare the DEGs list with each TF KO based on results from the literature and the relationship of the DEG to binding sites of the KO-TFs. To establish the binding sites of Atf3, the following datasets were used from the ENCODE portal (The ENCODE Project Consortium et al., 2012; Hitz et al., 2023 preprint; Luo et al., 2020) (https://www.encodeproject.org/) with the following identifiers: ENCSR205FOW, ENCSR480LIS, ENCSR879YNX, ENCSR000BUG, ENCSR000BKC, ENCSR000BJY, ENCSR402ZCY, ENCSR000BKE, ENCSR568ZXG, ENCSR632DCH, ENCSR000BNU, ENCSR000DOG and ENCSR028UIU (Table S8A). Peaks of these data sets were uploaded to the Galaxy web platform (https://usegalaxy.org/) (The Galaxy Community et al., 2024) and combined (~210,700 peaks) then merged with the mergeBED function of the bedtools (~97K peaks) (Quinlan and Hall, 2010). These coordinates were converted from human hg38 to mouse mm10 using UCSC Genome Browser-LiftOver (Hinrichs et al., 2006) with default settings for conversion across the species. Additionally, the ChIP-seq data in a mouse with mm9 (Zhang et al., 2023) were converted to mm10 with LiftOver using default parameters for the same species conversion. Later, all converted data were combined and merged, resulting in 65,307 peaks in mm10. As a next step, Atf3 peaks in mm10 were intersected with ATAC-seq data (R.C. and F.G., unpublished), which covers chromatin accessible data from D3 to D7 in whole EB with the bedtools 'Intersect interval' function with a minimum overlap fraction of 0.1, which resulted in 28,136 peaks.

These peaks were checked to determine whether or not they contain Atf3 motifs with the intersection of the following process: human and mouse motifs were selected from Jaspar (Castro-Mondragon et al., 2021; Fornes et al., 2019; Khan et al., 2017; Rauluseviciute et al., 2024), and MA605.1, MA0605.2, MA0605.3, MA1988.1, MA1988.2 and FIMO (Grant et al., 2011) were used for each motif on ATAC-seq peaks with an output threshold of 0.01, including reverse complement strand parameters. The resulting motifs were merged later. This merging resulted in 27,943 peaks in mm10, which were used to link them to the nearest gene with the GREAT web tool (http://great.stanford.edu/public/html/) (McLean et al., 2010; Tanigawa et al., 2022), which has a single nearest gene option within a 1000 kb distance. The output of this process was then used to overlap it with the DEG list of $\Delta Atf3$. In addition, the ChIPseeker package (Wang et al., 2022; Yu et al., 2015) was used for the peak distance to the TSS with hg38 and mm10 Ensembl genomic annotations (Harrison et al., 2024).

Harmonize 3.0 ENCODE Transcription Factor Targets database (Rouillard et al., 2016) was used for ATF3 targets. The mouse orthologs are converted from human with syngoportal-version-12-12-2020 (Koopmans et al., 2019) and intersected with the DEGs list of the $\Delta Atf3$.

Furthermore, five studies of the knockout and knockdown ATF3 in humans and mice were later intersected with the DEGs list (Badu et al.,

2024; Labzin et al., 2015; Di Marcantonio et al., 2021; Sood et al., 2017; Xu et al., 2011). A similar process was followed for *Zfp711* as described for *Atf3*. ChIP-seq data were used from GSE102616, GSE20673 and GSE145160 (Kleine-Kohlbrecher et al., 2010; Ni et al., 2020; Rhie et al., 2018). The motif of the ZNF711 has been described previously (Ni et al., 2020). For ZNF711 literature DEGs, GSE102616 and GSE145160 (Ni et al., 2020; Rhie et al., 2018; Snabel et al., 2025).

## Declaration of generative AI in the writing process
During the preparation of this work, the author(s) used Claude.ai and ChatGPT in order to improve clarity of writing. After using this tool/service, the author(s) reviewed and edited the content as needed, and take full responsibility for the content of the publication.

### Acknowledgements
We are grateful to Alex Maas and Tsung Wai Kan for their assistance with running the different samples in the FACS facility, to Cengizhan Acikel for statistical advice, and to Dilek Akyol for the financial support of R.C.

### Competing interests
The authors declare no competing or financial interests.

### Author contributions
Conceptualization: R.C., F.G.; Data curation: R.C., G.P., J.v.S., E.B., Y.F., F.G.; Formal analysis: R.C., R.H., G.v.B., M.A.S., A.K.; Investigation: R.C., F.G.; Methodology: R.C., R.H.; Project administration: R.C., F.G.; Resources: R.C., E.B.; Software: R.C., R.H., G.v.B., M.A.S., A.K.; Supervision: R.C., J.G., F.G.; Validation: R.C.; Visualization: R.C.; Writing – original draft: R.C., G.P., J.v.S., J.v.H., D.H., E.M., F.G.; Writing – review & editing: R.C., G.P., J.v.S., E.B., J.v.H., D.H., E.M., F.G.

### Funding
 Deposited in PMC for immediate release.

### Data and resource availability
scRNA-Seq data generated in this study have been deposited in the BioStudies database under the accession number E-MTAB-14678 and E-MTAB-14678 (Sarkans et al., 2017). UCSC browser sessions are accessible with the following links https://genome.ucsc.edu/s/mdrcetin/hg38_ATF3, https://genome.ucsc.edu/s/mdrcetin/mm10_Atf3, https://genome.ucsc.edu/s/mdrcetin/mm10_Zfp711, https://genome.ucsc.edu/s/mdrcetin/hg19_ZNF711 and https://genome.ucsc.edu/s/mdrcetin/hg38_ZNF711. The code used is available at https://github.io/mdrcetin/Cmo_Atf3_Zfp711_Bcl6b and https://ridvan-cetin.github.io/CMO_Atf3_Zfp711_Bcl6b/. Our scRNAseq can be explored with shinnyapp-link at https://ridvan-cetin.github.io/CMO_Atf3_Zfp711_Bcl6b/. All other relevant data and details of resources can be found within the article and its supplementary information.

### Peer review history
The peer review history is available online at https://journals.biologists.com/dev/lookup/doi/10.1242/dev.204792.reviewer-comments.pdf

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
