## [Peer Review File · Development (Cambridge, England)]

Distinct roles of *Atf3*, *Zfp711* and *Bcl6b* in early embryonic hematopoietic and endothelial lineage specification

Ridvan Cetin, Giulia Picco, Jente van Staalduinen, Eric Bindels, Remco Hoogenboezem, Gregory van Beek, Mathijs A. Sanders, Yaren Fidan, Ahmet Korkmaz, Joost Gribnau, Jeffrey van Haren, Danny Huylebroeck, Eskeatnaf Mulugeta and Frank Grosveld

DOI: 10.1242/dev.204792

Editor: Samantha A Morris

Review timeline

Original submission:	17 March 2025
Editorial decision:	12 May 2025
First revision received:	5 September 2025
Editorial decision:	27 October 2025
Second revision received:	30 October 2025
Accepted:	31 October 2025

Original submission

First decision letter

MS ID#: dev.204792

MS TITLE: Distinct Roles of *Atf3*, *Zfp711*, and *Bcl6b* in Early Embryonic Hematopoietic and Endothelial Lineage Specification

AUTHORS: Ridvan Cetin, Giulia Picco, Jente van Staalduinen, Eric Bindels, Remco Hoogenboezem, Gregory van Beek, Mathijs A. Sanders, Yaren Fidan, Ahmet Korkmaz, Joost Gribnau, Jeffrey van Haren, Danny Huylebroeck, Eskeatnaf Mulugeta and Frank Grosveld

Dear Dr Cetin,

I have now received all the referees' reports on the above manuscript, and have reached a decision. The referees' comments are appended below, or you can access them online: please go to:

As you will see, the referees express considerable interest in your work, but have some significant criticisms and recommend a substantial revision of your manuscript before we can consider publication. If you are able to revise the manuscript along the lines suggested, which may involve further experiments, I will be happy to receive a revised version of the manuscript. Your revised paper will be re-reviewed by one or more of the original referees, and acceptance of your manuscript will depend on your addressing satisfactorily the reviewers' major concerns. Please also note that Development will normally permit only one round of major revision. If it would be helpful, you are welcome to contact us to discuss your revision in greater detail. Please send us a point-by-point response indicating your plans for addressing the referees' comments, and we will look over this and provide further guidance.

Please attend to all of the reviewers' comments and ensure that you clearly highlight all changes made in the revised manuscript. Please avoid using 'Tracked changes' in Word files as these are lost in PDF conversion. I should be grateful if you would also provide a point-by-point response detailing how you have dealt with the points raised by the reviewers in the 'Response to Reviewers' box. If you do not agree with any of their criticisms or suggestions please explain clearly why this is so.

Reviewer 1*Advance summary and potential significance to field*

In this study, the authors explore the roles of three transcription factors (Atf3, Zfp711, and Bcl6b) in hematovascular development by combining in vitro differentiation of mouse embryonic stem cells with flow cytometry and single-cell RNA sequencing (scRNA-seq). Profiling embryoid bodies (EBs) at days 4 and 7, they aim to (1) define the transcription factors' roles in early mesoderm versus hematoendothelial lineage decisions, (2) assess their function at later stages such as the endothelial-to-hematopoietic transition (EHT), and (3) identify disrupted downstream pathways upon gene knockout. The dataset is technically robust, incorporating multiple knockout clones and replicate differentiations. The use of Milo for differential abundance analysis is appropriate, though the observed shifts in cell type proportions following Atf3 and Zfp711 deletion are modest.

However, the study lacks mechanistic insight into how these transcription factors regulate lineage progression, in part because the dynamics of lineage decisions are not effectively taken into account. The biological interpretation would be greatly strengthened by integrating the in vitro data with in vivo developmental atlases such as Pijuan-Sala et al., 2019, particularly to contextualize the "late mesoderm" cluster, which is central to the authors' claims but remains poorly defined. Mapping the EB-derived cells onto such a reference could clarify which developmental stages are represented and support analysis of lineage dynamics through pseudotime.

While Atf3 and Zfp711 are broadly expressed in the in vitro late mesoderm cluster, their expression in vivo is more restricted to endothelial and haematoendothelial progenitors arising from nascent mesoderm. The authors do not reconcile this discrepancy or provide temporal expression data for these factors in their differentiation system. Without trajectory analysis or a clear timeline of gene expression, it is difficult to interpret how these transcription factors influence early versus late developmental decisions.

Finally, the manuscript would benefit from a more focused presentation. Reducing the number of figures and centering the narrative on the molecular timing and lineage-specific roles of these transcription factors - anchored in developmental context - would enhance both clarity and impact. Better characterisation of key populations, especially the late mesoderm cluster, is essential to establish the relevance of the findings to normal hematovascular development.

Comments for the author

1. Integrate with in vivo scRNA-seq references: The manuscript would benefit significantly from integration with an in vivo reference atlas of mouse gastrulation and early organogenesis, such as the Pijuan-Sala et al., 2019 dataset, which the authors already cite. This would help to place the EB-derived populations into a developmental context and clarify how faithfully the in vitro system recapitulates embryonic differentiation trajectories.

2. Define 'late mesoderm' populations more clearly: The population labelled as 'late mesoderm' in the manuscript is not well defined. Integration with in vivo atlases, followed by label transfer, could help resolve its identity. For example, some of these cells express Tbx4 and posterior Hox genes, which may correspond to developing allantoic mesoderm populations. This is consistent with findings from a recent preprint (Theeuwes et al., 2024), which performs scRNA-seq on EBs and highlights allantoic-like cells with similar transcriptional signatures.

3. Clarify expression timing of transcription factors: The manuscript does not provide clear information on when Atf3, Zfp711, and Bcl6b are expressed during differentiation. Including a kinetic analysis of their expression kinetics - both in vitro and in vivo - would help identify when these TFs may act and in which lineages. This is particularly important if their effects are indirect or stage-specific.

4. Improve flow cytometry presentation: The flow cytometry data are difficult to interpret in their current form. The main figures should include both summary dot plots (which are currently in the

supplementary figures) and representative contour plots, allowing readers to more easily visually assess population changes. At present, replicate-level data are relegated to supplementary figures, and statistical comparisons are not clearly presented. Given that some of the observed changes are subtle, better visualization and statistical annotation are essential.

5. EMP gating strategy: The manuscript defines EMPs as CD45⁺ cells, but this deviates from established markers. EMPs are typically defined as CD41⁺, c-Kit⁺, CD16/32⁺ and express low levels of CD45, as established in McGrath et al. 2015. The current definition risks misclassifying other hematopoietic populations and should be revisited considering the literature.

6. Simplify and clarify cell type labels: The manuscript's interpretation is hindered by complex and non-standard cell type labels (Figure 3), often including clustering indices, cycle phase, and arbitrary marker genes. A clearer and more biologically interpretable naming scheme would greatly improve readability.

7. Refocus and streamline the manuscript: The manuscript is long and, at times, difficult to navigate. Many analyses are included, but the central biological story is obscured. A more focused narrative - highlighting key findings and their implications - would make the manuscript more impactful.

8. Consider downstream functional validation: The study would be strengthened by functional assays to probe the role of the transcription factors in lineage decisions, such as Atf3 in EMP differentiation. Currently, the work remains descriptive. Perturbation-rescue experiments or additional orthogonal validation experiments, such as haematopoietic colony forming assays, would add credence to the claims that EMPs are perturbed by the Atf3 mutation for example and what downstream impact is critical for this process.

Reviewer 2

Advance summary and potential significance to field

This manuscript describes the results of a thorough examination of the impacts of three genes encoding transcription factors (Atf3, Zfp11, and Bcl6b) on early stages of hematopoietic and endothelial differentiation. These three genes were chosen because they are upregulated at a key stage, but little is known about their role in these differentiation processes. The strategy was to knock out each of the three genes using CRISPR-cas9 techniques in mouse embryonic stem cells (mESCs) and measure the impacts on numbers of specific cell types (using flow cytometric analysis) and transcriptomes (using RNA-seq) as the mESCs differentiated to mesoderm, endothelium and blood cells, in a system that models the three waves of early hematopoiesis and endothelial differentiation. The experimental design used a multiplexed single cell analysis of the cell markers and transcriptomes. The multiplexing allowed mESC clones with the three mutated genes (one mutated gene per clone) to be analyzed together, thereby reducing batch effects. The resulting data were analyzed thoroughly using appropriate methods. The results were complex and are described in great detail. In brief, the results led to the conclusion that Atf3 has a clear role in increasing late mesodermal lineages and regulating EMPs at later stages. Zfp11 has a role in certain mesodermal and endothelial lineages, and it down-regulates the Atf3 gene. No significant phenotypes for the knockout of the Bcl6b gene were observed. The results of these experiments provide important new insights into the roles of these transcription factors, and the data provide useful resources for further work.

Comments for the author

The manuscript could be improved by addressing the following points.

(1) The text of the manuscript is comprehensive and detailed, but some aspects are redundant. Overall, the manuscript is very long and a reader can lose the main thread as they go through the detailed presentation. An effort should be made to streamline the manuscript to maintain a focus on the key conclusions.

(2) It would be helpful to add a comment on the use of in vitro differentiation of mESCs to mimic developmental steps in vivo. In the Conclusion (page 29, lines 696-697), it is stated that the current results lend further support to the ability of the in vitro system to mimic in vivo events, but it would be helpful to include some references in the Introduction as to the appropriateness of the system used.

(3) Conclusions should be couched in terms consistent with the in vitro system utilized. For example, p. 6, line 146, use "in an in vitro differentiation model of early embryonic hematopoietic development" instead of "in early embryonic hematopoietic development".

(4) p. 8, line 196: The statement "Bcl6b is expressed almost exclusively in mESC-derived HEPs" is too narrow, since Fig. 1D and E show substantial expression in endothelium also.

(5) page 10: In the section titled "Deletion of ATF3 shows an increased expression that parallels the progression to EHT", the information on expression of the Atf3 gene seems to focus on the patterns observed in the wild-type cells. It is not clear in this section what the effects on gene expression are upon deletion of the Atf3 gene. That material is presented in a later section. The title of the this section should be clarified.

(6) p. 14, line 330: Something seems missing in this sentence "Even though the EMP cell number is lower, the increase in DEGs seen in EMP is therefore a biological."

(7) p. 15, lines 364-368: Three different types of epigenetic data were used with respect to binding sites for ATF3, but it is not clear how they were used to define "Atf3 binding site(s)" close to DEG. It is important to distinguish occupancy by the transcription factor demonstrated by CHIP-seq compared to motif occurrences or motifs within the accessible chromatin (ATAC-seq data). A later section on the ZFP11 transcription factor says that regions positive for all three assays were used as binding sites (page 20, lines 503-507). Perhaps a similar approach was used for ATF3, but this issue should be clarified.

(8) line 459: Figure 3, not Figure 23.

(9) Figure 10 title: Perhaps the verb should be "subjected to" or "undergoing" rather than "submitted".

(10) page 24, lines 572 to 577: It is not clear what the term "ambient RNA" refers to, and this should be clarified. Perhaps it refers to RNA that is present only transiently in a cell, but if that is the case, is the detection of transient transcripts a problem or a feature?

(11) page 24, line 580: The term "epistatic" is ambiguous in this context. It would be more clear to just state that the upregulated genes are located in the genome downstream from (and maybe close by?) the mutated genes.

First revision

Author response to reviewers' comments

REVIEWER 1

We thank this reviewer for her/his very constructive comments.

- 1. Integrate with in vivo scRNA-seq references: The manuscript would benefit significantly from integration with an in vivo reference atlas of mouse gastrulation and early organogenesis, such as the Pijuan-Sala et al., 2019 dataset, which the authors already cite. This would help to place the EB-derived populations into a developmental context and clarify how faithfully the in vitro system recapitulates embryonic differentiation*

trajectories.

We have integrated our scRNA results with the literature and the mouse atlas (Imaz and Rosshandler 2023) in the results section and added supplementary figures S11, S22 and S30 to illustrate the points.

2. *Define 'late mesoderm' populations more clearly: The population labelled as 'late mesoderm' in the manuscript is not well defined. Integration with in vivo atlases, followed by label transfer, could help resolve its identity. For example, some of these cells express Tbx4 and posterior Hox genes, which may correspond to developing allantoic mesoderm populations. This is consistent with findings from a recent preprint (Theeuwes et al., 2024), which performs scRNA-seq on EBs and highlights allantoic-like cells with similar transcriptional signatures.*

We followed this advice. We mapped our data to the reference dataset (Imaz and Rosshandler 2023) and relabeled cell types in the late mesoderm, including a set of allantois like cells, which are also used by Theeuwes et al. 2025). We also addressed the issue of the (apparent) difference between the *in vitro* and *in vivo* expression of Atf3 and Zfp711 in the text and included this in Fig. S11 and S22. For example for ATF3 there is no discrepancy in endothelium, while in EMP there is agreement for the *in vitro* EMP#1 and the *in vivo* data, but the increase in expression in EMP#2 and #3 is not seen in the *in vivo* data. This is because the *in vivo* data only represent EMP#1 as demonstrated by the expression of e.g. *Kit*, *Adgrg1*, *Ctla2a*, *Hapln1* or *Nrgn* which are on in both EMP#1 and the *in vivo* data, but shut off in EMP#2 and #3. We conclude that the discrepancy is only apparent, because the *in vivo* data only represent the early stage EMP#1 (and perhaps early EMP#2). See Figs. S11 and S22.

3. Clarify expression timing of transcription factors: The manuscript does not provide clear information on when *Atf3*, *Zfp711*, and *Bcl6b* are expressed during differentiation. Including a kinetic analysis of their expression kinetics – both *in vitro* and *in vivo* – would help identify when these TFs may act and in which lineages. This is particularly important if their effects are indirect or stage-specific.

We have included a pseudotime analysis *in vitro* and *in vivo* (ATF3 Figs. S11, 12, 13; ZFP711 Figs. S22, S23, S24; Bcl6b figs S30, S31, S32). We also show at which embryonic stage, anatomical location and somite counts these genes are expressed (Figs. S11, S22, S30).

4. Improve flow cytometry presentation: The flow cytometry data are difficult to interpret in their current form. The main figures should include both summary dot plots (which are currently in the supplementary figures) and representative contour plots, allowing readers to more easily visually assess population changes. At present, replicate-level data are

relegated to supplementary figures, and statistical comparisons are not clearly presented. Given that some of the observed changes are subtle, better visualization and statistical annotation are essential.

We implemented these suggested changes (Figs 3, 6 and S36)

- 5. EMP gating strategy: The manuscript defines EMPs as CD45+ cells, but this deviates from established markers. EMPs are typically defined as CD41⁺, c-Kit⁺, CD16/32⁺ and express low levels of CD45, as established in McGrath et al. 2015. The current definition risks misclassifying other hematopoietic populations and should be revisited considering the literature.*

We have addressed the CD45 issue in Fig. S5 and we highlight this issue in the text. This CD45 expression being unique is true for this specific developmental window and not for other windows as suggested by the reviewer.

- 6. Simplify and clarify cell type labels: The manuscript's interpretation is hindered by complex and non-standard cell type labels (Figure 3), often including clustering indices, cycle phase, and arbitrary marker genes. A clearer and more biologically interpretable naming scheme would greatly improve readability.*

We have addressed the labeling (Figs.2 and S6) according to the results of the *in vivo* reference mapping.

- 7. Refocus and streamline the manuscript: The manuscript is long and, at times, difficult to navigate. Many analyses are included, but the central biological story is obscured. A more focused narrative – highlighting key findings and their implications – would make the manuscript more impactful.*

We have rewritten the manuscript by shortening, restructuring and changing the order of the topics.

- 8. Consider downstream functional validation: The study would be strengthened by functional assays to probe the role of the transcription factors in lineage decisions, such as Atf3 in EMP differentiation. Currently, the work remains descriptive. Perturbation-rescue experiments or additional orthogonal validation experiments, such as haematopoietic colony forming assays, would add credence to the claims that EMPs are perturbed by the Atf3 mutation for example and what downstream impact is critical for this process.*

We are not in a position to follow these suggestions about validation, which we would have loved to do. However the powers that be have regrettably closed the department and stopped all funds to do extra work due to severe budget cuts. We did use new available data to pursue the question of function and potential mechanism, which uncovered a central role of Spi1 and Cebpb binding with ATF3 during the EMP stage when they are activated regulating many newly expressed genes. This is now part of the results and discussion sections including Figs. 9A and S21).

REVIEWER 2:

We are grateful for the comments and suggestion by this reviewer.

SUGGESTIONS TO AUTHORS

The manuscript could be improved by addressing the following points.

- (1) The text of the manuscript is comprehensive and detailed, but some aspects are redundant. Overall, the manuscript is very long and a reader can lose the main thread as they go through the detailed presentation. An effort should be made to streamline the manuscript to maintain a focus on the key conclusions.*

We have rewritten the manuscript by shortening, restructuring and changing the order of

the topics.

(2) It would be helpful to add a comment on the use of in vitro differentiation of mESCs to mimic developmental steps in vivo. In the Conclusion (page 29, lines 696-697), it is stated that the current results lend further support to the ability of the in vitro system to mimic in vivo events, but it would be helpful to include some references in the Introduction as to the appropriateness of the system used.

We have added the relevant references to the Introduction.

(3) Conclusions should be couched in terms consistent with the in vitro system utilized. For example, p. 6, line 146, use "in an in vitro differentiation model of early embryonic hematopoietic development" instead of "in early embryonic hematopoietic development".

We have implemented this suggestion in the rewriting of the manuscript.

(4) p. 8, line 196: The statement "Bcl6b is expressed almost exclusively in mESC-derived HEPs" is too narrow, since

Fig. 1D and E show substantial expression in endothelium also.

The reviewer is correct and we have changed the statement (line 89 and 208 Figs. 1D,E; S1-S2 and Figs S30 to S32).

(5) page 10: In the section titled "Deletion of ATF3 shows an increased expression that parallels the progression to EHT", the information on expression of the Atf3 gene seems to focus on the patterns observed in the wild-type cells. It is not clear in this section what the effects on gene expression are upon deletion of the Atf3 gene. That material is presented in a later section. The title of the this section should be clarified.

We have addressed this issue in the rewritten the manuscript.

(6) p. 14, line 330: Something seems missing in this sentence "Even though the EMP cell number is lower, the increase in DEGs seen in EMP is therefore a biological."

We have addressed this issue in the rewritten the manuscript.

(7) p. 15, lines 364-368: Three different types of epigenetic data were used with respect to binding sites for ATF3, but it is not clear how they were used to define "Atf3 binding site(s)" close to DEG. It is important to distinguish occupancy by the transcription factor demonstrated by ChIP-seq compared to motif occurrences or motifs within the accessible chromatin (ATAC-seq data). A later section on the ZFP11 transcription factor says that regions positive for all three assays were used as binding sites (page 20, lines 503-507). Perhaps a similar approach was used for ATF3, but this issue should be clarified.

We have addressed this in the text of the results and Figs.S17, Atf3 and S27, Zfp711. We would have loved to do extra experiments but we are not in a position to do this. The powers that be have regrettably closed the department and stopped all funds to do extra work due to severe cuts in the faculty of Medicine.

(8) line 459: Figure 3, not Figure 23.

This was corrected.

(9) Figure 10 title: Perhaps the verb should be "subjected to" or "undergoing" rather than "submitted".

This is now Figure 9 with a new title: Summary of Atf3 and Zfp711 KO effects on mESC differentiation in the *in vitro* hemato-endothelial differentiation.

(10) page 24, lines 572 to 577: It is not clear what the term "ambient RNA" refers to, and this should be clarified. Perhaps it refers to RNA that is present only transiently in a cell, but if that is the case, is the detection of transient transcripts a problem or a feature?

This is addressed in the Materials and Methods section (line 421) where the term ambient RNA is defined.

(11) page 24, line 580: The term "epistatic" is ambiguous in this context. It would be more clear to just state that the upregulated genes are located in the genome downstream from (and maybe close by?) the mutated genes.

We no longer use the term "epistatic".

Second decision letter

MS ID#: dev.204792R1

MS TITLE: Distinct Roles of Atf3, Zfp711, and Bcl6b in Early Embryonic Hematopoietic and Endothelial Lineage Specification

AUTHORS: Ridvan Cetin, Giulia Picco, Jente van Staalduinen, Eric Bindels, Remco Hoogenboezem, Gregory van Beek, Mathijs A. Sanders, Yaren Fidan, Ahmet Korkmaz, Joost Gribnau, Jeffrey van Haren, Danny Huylebroeck, Eskeatnaf Mulugeta and Frank Grosveld

Dear Dr Cetin,

I have now received all the referees reports on the above manuscript, and have reached a decision. The referees' comments are appended below.

The overall evaluation is positive, and we would like to publish a revised version of your manuscript in Development, provided that the referees' comments can be satisfactorily addressed. Please attend to all of the reviewers' comments in your revised manuscript and detail your responses in a point-by-point reply. If you do not agree with any of their criticisms or suggestions, please explain clearly why this is so. If it would be helpful, you are welcome to contact us to discuss your revision in greater detail. Please send us a point-by-point response indicating your plans for addressing the referees' comments, and we will look over this and provide further guidance.

In addition to the reviewers' feedback, please make the following minor revisions:

- Figure 1C (and related schematics): The current diagram depicts a sequential progression from epiblast→ primitive streak→ ectoderm/mesoderm/PGCs, which is not biologically accurate. Please revise to indicate that some epiblast cells ingress through the primitive streak to form mesoderm and endoderm, while others remain as epiblast to become ectoderm.
- Figure presentation (optional): If possible, improve figure clarity and contrast to enhance visual readability. This is not a requirement for acceptance but would improve the presentation quality.

Reviewer 2

Advance summary and potential significance to field

This manuscript describes the results of a thorough examination of the impacts of three genes encoding transcription factors (Atf3, Zfp11, and Bcl6b) on early stages of hematopoietic and endothelial differentiation. In brief, the results led to the conclusion that Atf3 has a clear role in

increasing late mesodermal lineages and regulating EMPs at later stages. Zfp11 has a role in certain mesodermal and endothelial lineages, and it down-regulates the Atf3 gene. No significant phenotypes for the knockout of the Bcl6b gene were observed. The results of these experiments provide important new insights into the roles of these transcription factors, and the data provide useful resources for further work.

Comments for the author

The manuscript has been revised and extensively re-written. All concerns raised in my initial review have been thoroughly addressed. The revised manuscript is a concise and compelling report of important new data.

Second revision

Author response to reviewers' comments

In addition to the reviewers' feedback, please make the following minor revisions:

1. **Figure 1C (and related schematics):** *The current diagram depicts a sequential progression from epiblast → primitive streak → ectoderm/mesoderm/PGCs, which is not biologically accurate. Please revise to indicate that some epiblast cells ingress through the primitive streak to form mesoderm and endoderm, while others remain as epiblast to become ectoderm.*

We are grateful for the highlighting of this biological inaccuracy. We corrected Figure 1C and its derivatives that also present in Figure 9A,B.

2. **Figure presentation (optional):** *If possible, improve figure clarity and contrast to enhance visual readability. This is not a requirement for acceptance but would improve the presentation quality.*

We are grateful for these comments. We improved figure clarity, contrast and readability of Figure 1 to Figure 9.

REVIEWER 2

Reviewer 2: SUMMARY OF THE ADVANCE MADE IN THIS PAPER AND ITS POTENTIAL SIGNIFICANCE TO THE FIELD

This manuscript describes the results of a thorough examination of the impacts of three genes encoding transcription factors (Atf3, Zfp11, and Bcl6b) on early stages of hematopoietic and endothelial differentiation. In brief, the results led to the conclusion that Atf3 has a clear role in increasing late mesodermal lineages and regulating EMPs at later stages. Zfp11 has a role in certain mesodermal and endothelial lineages, and it down-regulates the Atf3 gene. No significant phenotypes for the knockout of the Bcl6b gene were observed. The results of these experiments provide important new insights into the roles of these transcription factors, and the data provide useful resources for further work.

SUGGESTIONS TO AUTHORS

The manuscript has been revised and extensively re-written. All concerns raised in my initial review have been thoroughly addressed. The revised manuscript is a concise and compelling report of important new data.

We are grateful for the comments and suggestions by this reviewer.

REVIEWER 1

We thank this reviewer for her/his very constructive comments.

1. *Integrate with in vivo scRNA-seq references: The manuscript would benefit significantly from integration with an in vivo reference atlas of mouse gastrulation and early organogenesis, such as the Pijuan-Sala et al., 2019 dataset, which the authors already cite. This would help to place the EB-derived populations into a developmental context and clarify how faithfully the in vitro system recapitulates embryonic differentiation trajectories.*

We have integrated our scRNA results with the literature and the mouse atlas (Imaz and Rosshandler 2023) in the results section and added supplementary figures S11, S22 and S30 to illustrate the points.

2. *Define 'late mesoderm' populations more clearly: The population labelled as 'late mesoderm' in the manuscript is not well defined. Integration with in vivo atlases, followed by label transfer, could help resolve its identity. For example, some of these cells express Tbx4 and posterior Hox genes, which may correspond to developing allantoic mesoderm populations. This is consistent with findings from a recent preprint (Theeuwes et al., 2024), which performs scRNA-seq on EBs and highlights allantoic-like cells with similar transcriptional signatures.*

We followed this advice. We mapped our data to the reference dataset (Imaz and Rosshandler 2023) and relabeled cell types in the late mesoderm, including a set of allantois like cells, which are also used by Theeuwes et al. 2025). We also addressed the issue of the (apparent) difference between the *in vitro* and *in vivo* expression of Atf3 and Zfp711 in the text and included this in Fig. S11 and S22. For example for ATF3 there is no discrepancy in endothelium, while in EMP there is agreement for the *in vitro* EMP#1 and the *in vivo* data, but the increase in expression in EMP#2 and #3 is not seen in the *in vivo* data. This is because the *in vivo* data only represent EMP#1 as demonstrated by the expression of e.g. *Kit*, *Adgrg1*, *Ctla2a*, *Hapln1* or *Nrgn* which are on in both EMP#1 and the *in vivo* data, but shut off in EMP#2 and #3. We conclude that the discrepancy is only apparent, because the *in vivo* data only represent the early stage EMP#1 (and perhaps early EMP#2). See Figs. S11 and S22.

3. Clarify expression timing of transcription factors: The manuscript does not provide clear information on when *Atf3*, *Zfp711*, and *Bcl6b* are expressed during differentiation. Including a kinetic analysis of their expression kinetics – both *in vitro* and *in vivo* – would help identify when these TFs may act and in which lineages. This is particularly important if their effects are indirect or stage-specific.

We have included a pseudotime analysis *in vitro* and *in vivo* (*ATF3* Figs. S11, 12, 13; *ZFP711* Figs. S22, S23, S24; *Bcl6b* figs S30, S31, S32). We also show at which embryonic stage, anatomical location and somite counts these genes are expressed (Figs. S11, S22, S30).

4. Improve flow cytometry presentation: The flow cytometry data are difficult to interpret in their current form. The main figures should include both summary dot plots (which are currently in the supplementary figures) and representative contour plots, allowing readers to more easily visually assess population changes. At present, replicate-level data are relegated to supplementary figures, and statistical comparisons are not clearly presented. Given that

some of the observed changes are subtle, better visualization and statistical annotation are essential.

We implemented these suggested changes (Figs 3, 6 and S36)

- 5. EMP gating strategy: The manuscript defines EMPs as CD45+ cells, but this deviates from established markers. EMPs are typically defined as CD41⁺, c-Kit⁺, CD16/32⁺ and express low levels of CD45, as established in McGrath et al. 2015. The current definition risks misclassifying other hematopoietic populations and should be revisited considering the literature.*

We have addressed the CD45 issue in Fig. S5 and we highlight this issue in the text. This CD45 expression being unique is true for this specific developmental window and not for other windows as suggested by the reviewer.

- 6. Simplify and clarify cell type labels: The manuscript's interpretation is hindered by complex and non-standard cell type labels (Figure 3), often including clustering indices, cycle phase, and arbitrary marker genes. A clearer and more biologically interpretable naming scheme would greatly improve readability.*

We have addressed the labeling (Figs.2 and S6) according to the results of the *in vivo* reference mapping.

- 7. Refocus and streamline the manuscript: The manuscript is long and, at times, difficult to navigate. Many analyses are included, but the central biological story is obscured. A more focused narrative – highlighting key findings and their implications – would make the manuscript more impactful.*

We have rewritten the manuscript by shortening, restructuring and changing the order of the topics.

- 8. Consider downstream functional validation: The study would be strengthened by functional assays to probe the role of the transcription factors in lineage decisions, such as Atf3 in EMP differentiation. Currently, the work remains descriptive. Perturbation-rescue experiments or additional orthogonal validation experiments, such as haematopoietic colony forming assays, would add credence to the claims that EMPs are perturbed by the Atf3 mutation for example and what downstream impact is critical for this process.*

We are not in a position to follow these suggestions about validation, which we would have loved to do. However the powers that be have regrettably closed the department and stopped all funds to do extra work due to severe budget cuts. We did use new available data to pursue the question of function and potential mechanism, which uncovered a central role of Spi1 and Cebpb binding with ATF3 during the EMP stage when they are activated regulating many newly expressed genes. This is now part of the results and discussion sections including Figs. 9A and S21).

REVIEWER 2:

We are grateful for the comments and suggestion by this reviewer.

SUGGESTIONS TO AUTHORS

The manuscript could be improved by addressing the following points.

- (1) The text of the manuscript is comprehensive and detailed, but some aspects are redundant. Overall, the manuscript is very long and a reader can lose the main thread as they go through the detailed presentation. An effort should be made to streamline the manuscript to maintain a focus on the key conclusions.*

We have rewritten the manuscript by shortening, restructuring and changing the order of the topics.

(2) It would be helpful to add a comment on the use of in vitro differentiation of mESCs to mimic developmental steps in vivo. In the Conclusion (page 29, lines 696-697), it is stated that the current results lend further support to the ability of the in vitro system to mimic in vivo events, but it would be helpful to include some references in the Introduction as to the appropriateness of the system used.

We have added the relevant references to the Introduction.

(3) Conclusions should be couched in terms consistent with the in vitro system utilized. For example, p. 6, line 146, use "in an in vitro differentiation model of early embryonic hematopoietic development" instead of "in early embryonic hematopoietic development".

We have implemented this suggestion in the rewriting of the manuscript.

(4) p. 8, line 196: The statement "Bcl6b is expressed almost exclusively in mESC-derived HEPs" is too narrow, since Fig. 1D and E show substantial expression in endothelium also.

The reviewer is correct and we have changed the statement (line 89 and 208 Figs. 1D,E; S1-S2 and Figs S30 to S32).

(5) page 10: In the section titled "Deletion of ATF3 shows an increased expression that parallels the progression to EHT", the information on expression of the Atf3 gene seems to focus on the patterns observed in the wild-type cells. It is not clear in this section what the effects on gene expression are upon deletion of the Atf3 gene. That material is presented in a later section. The title of this section should be clarified.

We have addressed this issue in the rewritten manuscript.

(6) p. 14, line 330: Something seems missing in this sentence "Even though the EMP cell number is lower, the increase in DEGs seen in EMP is therefore a biological."

We have addressed this issue in the rewritten manuscript.

(7) p. 15, lines 364-368: Three different types of epigenetic data were used with respect to binding sites for ATF3, but it is not clear how they were used to define "Atf3 binding site(s)" close to DEG. It is important to distinguish occupancy by the transcription factor demonstrated by ChIP-seq compared to motif occurrences or motifs within the accessible chromatin (ATAC-seq data). A later section on the ZFP11 transcription factor says that regions positive for all three assays were used as binding sites (page 20, lines 503-507). Perhaps a similar approach was used for ATF3, but this issue should be clarified.

We have addressed this in the text of the results and Figs.S17, Atf3 and S27, Zfp711. We would have loved to do extra experiments but we are not in a position to do this. The powers that be have regrettably closed the department and stopped all funds to do extra work due to severe cuts in the faculty of Medicine.

(8) line 459: Figure 3, not Figure 23.

This was corrected.

(9) Figure 10 title: Perhaps the verb should be "subjected to" or "undergoing" rather than "submitted".

This is now Figure 9 with a new title: Summary of Atf3 and Zfp711 KO effects on mESC differentiation in the *in vitro* hemato-endothelial differentiation.

(10) page 24, lines 572 to 577: It is not clear what the term "ambient RNA" refers to, and this should be clarified. Perhaps it refers to RNA that is present only transiently in a cell, but if that is the case, is the detection of transient transcripts a problem or a feature?

This is addressed in the Materials and Methods section (line 421) where the term ambient RNA is defined.

(11) page 24, line 580: The term "epistatic" is ambiguous in this context. It would be more clear to just state that the upregulated genes are located in the genome downstream from (and maybe close by?) the mutated genes.

We no longer use the term "epistatic".

Third decision letter

MS ID#: dev.204792R2

MS TITLE: Distinct Roles of Atf3, Zfp711, and Bcl6b in Early Embryonic Hematopoietic and Endothelial Lineage Specification

AUTHORS: Ridvan Cetin, Giulia Picco, Jente van Staalduinen, Eric Bindels, Remco Hoogenboezem, Gregory van Beek, Mathijs A. Sanders, Yaren Fidan, Ahmet Korkmaz, Joost Gribnau, Jeffrey van Haren, Danny Huylebroeck, Eskeatnaf Mulugeta and Frank Grosveld

Dear Dr Cetin,

I am happy to tell you that your manuscript has been accepted for publication in Development, pending our standard publication integrity checks.